This work is distributed under
the Creative Commons Attribution 4.0 License.

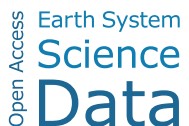

# Synoptic analysis of a decade of daily measurements of SO₂ emission in the troposphere from volcanoes of the global ground-based Network for Observation of Volcanic and Atmospheric Change

**Santiago Arellano[1], Bo Galle[1], Fredy Apaza[2], Geoffroy Avard[3], Charlotte Barrington[4], Nicole Bobrowski[5], Claudia Bucarey[6], Viviana Burbano[7,✝], Mike Burton[8,a], Zoraida Chacón[7], Gustavo Chigna[9], Christian Joseph Clarito[10], Vladimir Conde[1], Fidel Costa[4], Maarten De Moor[3], Hugo Delgado-Granados[11], Andrea Di Muro[12], Deborah Fernandez[10], Gustavo Garzón[7], Hendra Gunawan[13], Nia Haerani[13], Thor H. Hansteen[14], Silvana Hidalgo[15], Salvatore Inguaggiato[8], Mattias Johansson[1], Christoph Kern[16], Manne Kihlman[1], Philippe Kowalski[12], Pablo Masias[2], Francisco Montalvo[17], Joakim Möller[18], Ulrich Platt[5], Claudia Rivera[1,b], Armando Saballos[19], Giuseppe Salerno[8], Benoit Taisne[4], Freddy Vásconez[15], Gabriela Velásquez[6], Fabio Vita[8], and Mathieu Yalire[20]**

[1]Department of Space, Earth and Environment, Chalmers University of Technology CE1 TS1, Sweden
[2]Instituto Geológico, Minero y Metalúrgico (INGEMMET), TS2, Peru
[3]Observatorio Vulcanológico y Sismológico de Costa Rica (OVSICORI), TS3, Costa Rica
[4]Earth Observatory of Singapore (EOS), Nanyang Technological University, Singapore
[5]Institute of Environmental Physics, Heidelberg University, TS4, Germany
[6]Servicio Nacional de Geología y Minería (SERNAGEOMIN), TS5, Chile
[7]Servicio Geológico Colombiano (SGC), TS6, Colombia
[8]Istituto Nazionale di Geofisica e Vulcanologia (INGV), TS7, Italy
[9]Instituto Nacional de Sismología, Vulcanología, Meteorología e Hidrología (INSIVUMEH), TS8, Guatemala
[10]Philippine Institute of Volcanology and Seismology (PHIVOLCS), TS9, Philippines
[11]Instituto de Geofísica, Universidad Nacional Autónoma de México (UNAM), TS10, Mexico
[12]Observatoire Volcanologique du Piton de la Fournaise, Institut de Physique du Globe de Paris (IPGP), CE2 TS11, France
[13]Center for Volcanology and Geological Hazard Mitigation (CVGHM), TS12, Indonesia
[14]GEOMAR Helmholtz Centre for Ocean Research Kiel, Kiel TS13, Germany
[15]Instituto Geofísico (IGEPN), Escuela Politécnica Nacional, CE3 TS14, Ecuador
[16]Volcano Disaster Assistance Program (VDAP), United States Geological Survey (USGS), CE4 TS15, United States
[17]Servicio Nacional de Estudios Territoriales (SNET), TS16, El Salvador
[18]Möller Data Workflow Systems AB (MolFlow), TS17, Sweden
[19]Instituto Nicaragüense de Estudios Territoriales (INETER), TS18, Nicaragua
[20]Observatoire Volcanologique de Goma (OVG), TS19, Democratic Republic of the Congo CE5
[a]now at: School of Earth, Atmospheric and Environmental Sciences, CE6 University of Manchester, Manchester TS20, UK
[b]now at: Centro de Ciencias de la Atmósfera, Universidad Nacional Autónoma de México, CE7 TS21, Mexico
✝deceased

**Correspondence:** Santiago Arellano (santiago.arellano@chalmers.se)

Received: 1 October 2020 – Discussion started: 3 November 2020
Revised: 27 January 2021 – Accepted: 11 February 2021 – Published:

**Abstract.** `TS22` Volcanic plumes are common and far-reaching manifestations of volcanic activity during and between eruptions. Observations of the rate of emission and composition of volcanic plumes are essential to recognize and, in some cases, predict the state of volcanic activity. Measurements of the size and location of the plumes are important to assess the impact of the emission from sporadic or localized events to persistent or widespread processes of climatic and environmental importance. These observations provide information on volatile budgets on Earth, chemical evolution of magmas, and atmospheric circulation and dynamics. Space-based observations during the last decades have given us a global view of Earth's volcanic emission, particularly of sulfur dioxide ($SO_2$). Although none of the satellite missions were intended to be used for measurement of volcanic gas emission, specially adapted algorithms have produced time-averaged global emission budgets. These have confirmed that tropospheric plumes, produced from persistent degassing of weak sources, dominate the total emission of volcanic $SO_2$. Although space-based observations have provided this global insight into some aspects of Earth's volcanism, it still has important limitations. The magnitude and short-term variability of lower-atmosphere emissions, historically less accessible from space, remain largely uncertain. Operational monitoring of volcanic plumes, at scales relevant for adequate surveillance, has been facilitated through the use of ground-based scanning differential optical absorption spectrometer (ScanDOAS) instruments since the beginning of this century, largely due to the coordinated effort of the Network for Observation of Volcanic and Atmospheric Change (NOVAC). In this study, we present a compilation of results of post-analysis homogenized measurements of $SO_2$ flux and plume parameters obtained during the period March 2005 to January 2017 of 32 volcanoes in NOVAC. This inventory opens a window into the short-term emission patterns of a diverse set of volcanoes in terms of magma composition, geographical location, magnitude of emission, and style of eruptive activity. We find that passive volcanic degassing is by no means a stationary process in time and that large sub-daily variability is observed in the flux of volcanic gases, which has implications for emission budgets produced using short-term, sporadic observations. The use of a standard evaluation method allows for intercomparison between different volcanoes and between ground- and space-based measurements of the same volcanoes. The emission of several weakly degassing volcanoes, undetected by satellites, is presented for the first time. We also compare our results with those reported in the literature, providing ranges of variability in emission not accessible in the past. The open-access data repository introduced in this article will enable further exploitation of this unique dataset, with a focus on volcanological research, risk assessment, satellite-sensor validation, and improved quantification of the prevalent tropospheric component of global volcanic emission.

Datasets for each volcano are made available at https://novac.chalmers.se (last access: `TS23`) under the CC-BY 4 license or through the DOI (digital object identifier) links provided in Table 1.

# 1 Introduction

Volcanic eruptions are to a large extent triggered or modulated by the intricate dynamics of segregation and escape of volatiles from magmas, making the observation of the rate of gas emission an important component of monitoring efforts to identify and predict the state of a volcanic system (Sparks, 2003; Sparks et al., 2012). The resulting atmospheric plumes are the farthest-reaching products of volcanic activity and constitute rich environments for a number of important processes affecting the physics and chemistry of the atmosphere, the radiative balance of the climate system, or the biogeochemical impact on soils and the ocean (e.g. Robock, 2000; Langmann, 2014; Schmidt et al., 2018).

Volcanoes are sources of many trace atmospheric compounds, such as water vapour ($H_2O$), carbon dioxide ($CO_2$), sulfur dioxide ($SO_2$), carbonyl sulfide (OCS), hydrogen chloride (HCl), hydrogen fluoride (HF), hydrogen sulfide ($H_2S$), and molecular hydrogen ($H_2$), as well as solid particles and metals. From these species, $SO_2$ is the most widely observed by passive optical remote sensing methods (Oppenheimer, 2010). This is a consequence of its low atmospheric background and accessible radiation absorption bands, particularly in the near-ultraviolet (NUV) and mid-infrared (MIR) spectral regions. This is advantageous for several reasons, for example, for (1) the volcanologist, $SO_2$ is a reliable tracer of magmatic activity due to its strongly pressure-dependent solubility in magmas. Since $H_2O$ is usually the most abundant volatile species and thus the most important driver of volcanic activity and has a pressure-dependent solubility, both $H_2O$ and $SO_2$ fluxes are positively correlated with eruptive intensity. For (2) the climatologist, $SO_2$ may be transformed by a series of reactions into aerosols containing sulfuric acid ($H_2SO_4$), which exert a strong radiative forcing, especially when reaching the stratosphere. Or, for (3) the meteorologist, $SO_2$ has a long enough residence time in the atmosphere to serve as a tracer of volcanic plume transport at regional or even global scales.

Measurements of the mass emission rate or flux of $SO_2$ from volcanoes started in the 1970s with the development

**Table 1.** Statistics of measured SO$_2$ flux during 2005–2016 for 32 volcanoes in NOVAC (Network for Observation of Volcanic and Atmospheric Change). The reported values are the arithmetic mean (average) of the different daily statistics (mean; standard deviation, SD; and quartiles) included in the NOVAC data files. DOI: digital object identifier.

| Volcano | Statistics of measured SO$_2$ flux during 2005–2016 [kg/s] | | | | | Number of days with valid flux statistics with respect to days with measurements | Reference | DOI link |
|---|---|---|---|---|---|---|---|---|
| | Mean | SD | First quartile | Median | Third quartile | | | |
| Arenal | 1.59 | 0.80 | 1.04 | 1.43 | 1.98 | 38/350 | Avard et al. (2020a) | https://doi.org/10.17196/novac.arenal.001 |
| Concepción | 6.10 | 2.01 | 4.66 | 5.92 | 7.30 | 186/795 | Saballos et al. (2020a) | https://doi.org/10.17196/novac.concepcion.001 |
| Copahue | 9.02 | 3.61 | 6.51 | 8.37 | 11.03 | 288/476 | Velásquez et al. (2020a) | https://doi.org/10.17196/novac.copahue.001 |
| Cotopaxi | 8.94 | 5.31 | 5.24 | 8.38 | 11.76 | 447/2829 | Hidalgo et al. (2020a) | https://doi.org/10.17196/novac.cotopaxi.001 |
| Etna | 40.72 | 12.18 | 31.98 | 38.88 | 47.94 | 192/875 | Salerno et al. (2020) | https://doi.org/10.17196/novac.etna.001 |
| Fuego de Colima | 2.72 | 1.66 | 1.54 | 2.41 | 3.60 | 185/2750 | Delgado et al. (2020a) | https://doi.org/10.17196/novac.fuegodecolima.001 |
| Fuego | 3.53 | 1.30 | 2.61 | 3.41 | 4.24 | 30/407 | Chigna et al. (2020a) | https://doi.org/10.17196/novac.fuegoguatemala.001 |
| Galeras | 8.99 | 3.80 | 6.35 | 8.46 | 11.11 | 704/3340 | Chacón et al. (2020a) | https://doi.org/10.17196/novac.galeras.001 |
| Isluga | 8.11 | 3.65 | 5.46 | 7.35 | 10.18 | 230/497 | Bucarey et al. (2020a) | https://doi.org/10.17196/novac.isluga.001 |
| Lascar | 2.62 | 1.43 | 1.61 | 2.32 | 3.37 | 75/919 | Bucarey et al. (2020) TS24 | https://doi.org/10.17196/novac.nyiragongo.001 |
| Llaima | 10.08 | 3.42 | 7.66 | 10.08 | 12.49 | 4/1308 | Bucarey et al. (2020c) | https://doi.org/10.17196/novac.llaima.001 |
| Masaya | 3.87 | 1.38 | 2.91 | 3.74 | 4.70 | 500/772 | Saballos et al. (2020b) | https://doi.org/10.17196/novac.masaya.001 |
| Mayon | 7.01 | 2.50 | 5.30 | 6.59 | 8.29 | 102/1173 | Bornas et al. (2020) | https://doi.org/10.17196/novac.mayon.001 |
| Momotombo | 1.85 | 0.60 | 1.45 | 1.75 | 2.18 | 35/158 | Saballos et al. (2020c) | https://doi.org/10.17196/novac.momotombo.001 |
| Nevado del Ruiz | 7.76 | 4.96 | 4.23 | 6.91 | 10.43 | 1281/2431 | Chacón et al. (2020b) | https://doi.org/10.17196/novac.nevadodelruiz.001 |
| Nyiragongo | 19.14 | 8.89 | 12.71 | 17.71 | 24.52 | 432/1758 | Yalire et al. (2020) | https://doi.org/10.17196/novac.nyiragongo.001 |
| Piton de la Fournaise | 9.58 | 6.77 | 4.65 | 8.62 | 13.20 | 22/3402 | Di Muro et al. (2020) | https://doi.org/10.17196/novac.pitondelafournaise.001 |
| Planchón–Peteroa | 0.05 | 0.02 | 0.03 | 0.04 | 0.06 | 4/22 | Velásquez et al. (2020b) | https://doi.org/10.17196/novac.planchonpeteroa.001 |
| Popocatépetl | 24.48 | 12.28 | 15.81 | 23.27 | 31.96 | 1207/3306 | Delgado et al. (2020b) | https://doi.org/10.17196/novac.popocatepetl.001 |
| Sabancaya | 11.86 | 5.85 | 7.56 | 10.76 | 15.10 | 126/162 | Masias et al. (2020b) | https://doi.org/10.17196/novac.sabancaya.001 |
| San Cristóbal | 8.87 | 3.41 | 6.44 | 8.34 | 10.86 | 1028/1557 | Saballos et al. (2020d) | https://doi.org/10.17196/novac.sancristobal.001 |
| San Miguel | 22.51 | 6.17 | 19.42 | 22.25 | 25.25 | 4/160 | Montalvo et al. (2020a) | https://doi.org/10.17196/novac.sanmiguel.001 |
| Sangay | 7.49 | 3.34 | 4.87 | 7.15 | 9.73 | 7/536 | Hidalgo et al. (2020c) | https://doi.org/10.17196/novac.sangay.001 |
| Santa Ana | 1.96 | 0.56 | 1.55 | 1.86 | 2.36 | 22/868 | Montalvo et al. (2020b) | https://doi.org/10.17196/novac.santaana.001 |
| Santiaguito | 3.22 | 1.80 | 1.96 | 2.83 | 4.08 | 170/570 | Chigna et al. (2020b) | https://doi.org/10.17196/novac.santiaguito.001 |
| Sinabung | 4.42 | 2.07 | 3.05 | 3.95 | 5.14 | 108/173 | Kasbani et al. (2020) | https://doi.org/10.17196/novac.sinabung.001 |
| Telica | 0.83 | 0.31 | 0.60 | 0.78 | 1.00 | 205/460 | Saballos et al. (2020e) | https://doi.org/10.17196/novac.telica.001 |
| Tungurahua | 17.12 | 7.93 | 11.54 | 16.17 | 21.81 | 1100/3463 | Hidalgo et al. (2020b) | https://doi.org/10.17196/novac.tungurahua.001 |
| Turrialba | 11.50 | 4.98 | 7.98 | 10.96 | 14.48 | 614/1878 | Avard et al. (2020b) | https://doi.org/10.17196/novac.turrialba.001 |
| Ubinas | 3.52 | 2.00 | 2.09 | 3.20 | 4.62 | 375/622 | Masias et al. (2020a) | https://doi.org/10.17196/novac.ubinas.001 |
| Villarrica | 6.75 | 2.58 | 4.90 | 6.28 | 8.23 | 203/2001 | Velásquez et al. (2020c) | https://doi.org/10.17196/novac.villarrica.001 |
| Vulcano | 0.20 | 0.06 | 0.15 | 0.19 | 0.24 | 34/1180 | Vita et al. (2020) | https://doi.org/10.17196/novac.vulcano.001 |

and application of the correlation spectrometer (COSPEC) (Moffat and Millán, 1971; Stoiber et al., 1973 TS25). This instrument disperses ultraviolet sky radiation using a grating and employs a mechanical mask to correlate the intensity of diffused solar radiation in the near-ultraviolet region at selected narrow bands, matching absorption features of $SO_2$. With proper calibration using cells containing $SO_2$ at known concentrations, the COSPEC instrument measures the column density of $SO_2$ relative to background by the methods of differential absorption. Flux is quantified assuming mass conservation: the volcanic source emission strength is equal to the integrated flux across a surface surrounding the volcano when no other sources or sinks are enclosed. The integrated flux is measured by scanning through a surface perpendicular to plume transport, integrating the column densities in the plume cross section, and multiplying this integral by the corresponding transport speed. COSPEC was typically used for sporadic or periodical field surveys, during both volcanic crises and periods of passive degassing. The first global emission budgets for volcanic $SO_2$ were based on extrapolation of these sporadic measurements on a fraction of globally degassing volcanoes, through a series of non-verified assumptions regarding the statistics of emission for measured and non-measured sources. Halmer and Schmincke (2002) TS26 recognized this problem and highlighted the need for increasing (i) the number of monitored volcanoes, (ii) the periods of observation, (iii) the sampling frequency of the measurements, and (iv) the homogeneity of protocols of measurement by different observers.

In the late 1970s, the first satellite-based sensors, intended primarily for monitoring the stratospheric ozone ($O_3$) layer, opened up the possibility of mapping and quantifying volcanogenic $SO_2$ from space (Krueger, 1983; Krueger et al., 1995). The successful Total Ozone Mapping Spectrometer (TOMS) instrument programme was succeeded by a series of optical instruments such as the Global Ozone Monitoring Experiment (GOME/GOME-2), the Scanning Imaging Absorption Spectrometer for Atmospheric Cartography (SCHIA-MACHY), the Ozone Monitoring Instrument (OMI), and the Ozone Mapping and Profiler Suite (OMPS). Infrared (IR) sensors, such as Infrared Atmospheric Sounding Interferometer (IASI) or Moderate Resolution Imaging Spectroradiometer (MODIS), have been also used for routine global observation of volcanic emissions (Khokhar, 2005 TS27; Carn et al., 2013; Theys et al., 2013). More recently, the Tropospheric Monitoring Instrument (TROPOMI), on board ESA's Sentinel-5 Precursor satellite since 2017, achieves a factor of 3 to 4 better sensitivity than OMI, due to better spatial resolution and sensor performance. This makes detection of weak emissions of $SO_2$ in the lower atmosphere feasible every day and with global coverage. Under ideal measurement conditions and knowledge of plume velocity, time series of volcanic $SO_2$ flux as low as $\sim 1\,kg/s$ (for 1 m/s wind speed) with sub-daily frequency can be derived from TROPOMI (Queißer et al., 2019; Theys et al., 2019).

During the 1990s and early 2000s smaller, cheaper, and more accurate and versatile alternatives to the COSPEC instrument were developed, in particular the miniaturized differential optical absorption spectrometer (MiniDOAS CE8) (Galle et al., 2003). This instrument incorporates a grating spectrometer to obtain the spectrum of diffused solar radiation in the UV (ultraviolet) spectrum and retrieves the relative column density of $SO_2$ by the DOAS method (Platt and Stutz, 2008). This line of research led to the implementation of fully automated scanning DOAS (or ScanDOAS) systems (Edmonds et al., 2003), which have enabled volcanological observatories to conduct nearly continuous monitoring of volcanic plumes. A version of this instrument, known as dual-beam scanning DOAS, can measure the plume velocity, height, and the integrated $SO_2$ flux in near to real time, with a time resolution of 1–15 min during daylight hours (Johansson et al., 2009). Similar spectroscopic instruments have been developed or replicated by different groups (Horton et al., 2006; Mori et al., 2007; Arellano et al., 2008; Burton et al., 2008).

Among other methods for ground-based optical remote sensing of integrated volcanic flux we highlight different types of imaging systems such as an imaging DOAS (I-DOAS) (Bobrowski et al., 2006; Louban et al., 2009) and thermal imaging Fourier transform infra-red (FTIR) spectrometry (Stremme et al., 2012), as well as UV and IR $SO_2$ cameras based on broadband filters or interferometry (Mori and Burton, 2006; Bluth et al., 2007; Kern et al., 2010b; Kuhn et al., 2014; Prata and Bernardo, 2014; Platt et al., 2015; McGonigle et al., 2017; Smekens et al., 2018 TS28). A crucial advantage of these systems, compared with ScanDOAS systems, is their higher temporal resolution and accurate quantification of plume speed by image-correlation techniques. Among the disadvantages we mention are that they usually require more restricted measurement conditions with respect to measurement geometry and weather; have a higher susceptibility to interference (e.g. aerosols); are usually designed for measurement of a single species; and require calibration by another instrument, usually a MiniDOAS system.

An important step towards extending the newly available tools for permanent volcanic gas monitoring has been the creation of the Network for Observation of Volcanic and Atmospheric Change (NOVAC) in 2005. The network was established with funding from the European Union (EU) during 2005–2010, and it has continued and expanded with resources from volcanological observatories and cooperating research groups, the Deep Carbon Observatory programme (https://deepcarbon.net/, last access: TS29), the Volcano Disaster Assistance Program (VDAP) of the United States Geological Survey (USGS) and the United States Agency for International Development (USAID), and Chalmers University of Technology. The main purpose of the NOVAC project was to set up local monitoring networks of dual-beam Scan-DOAS instruments. It started with 15 volcanoes monitored by observatories in Latin America, the Democratic Republic

Please note the remarks at the end of the manuscript.

of the Congo, Reunion Island, and Italy, involving 18 different groups with expertise in volcanology, atmospheric remote sensing, and meteorology. At the time of writing, NOVAC has included about 160 stations at 47 volcanoes in different regions around the world, now including Iceland, the Philippines, Indonesia, Papua New Guinea, and Montserrat. The advantages of these instruments with respect to spaceborne sensors include continuous calibration, better temporal and spatial resolution, more direct measurement of flux, and better sensitivity to tropospheric plumes. A key disadvantage is the limited spatial coverage inherent to ground networks. Details of the instrument and operation routines are given in Galle et al. (2010). Figure 1 shows a map with locations of the volcanoes that have been part of NOVAC.

The purpose of this paper is to present an inventory of daily flux measurements of SO$_2$ obtained in NOVAC from 1 March 2005 until 31 January 2017. These results were obtained by standardized re-evaluation of the collected spectra, incorporating information about wind velocity from a re-analysis dataset provided by the European Centre for Medium-Range Weather Forecasts (ECMWF). We present daily statistics of emission and corresponding information about plume parameters. A database for access to the results is described in detail, providing a substantial basis for further investigations of volcanic degassing patterns over time. We compare the emission inventory of NOVAC with past compilations of degassing intensity on these volcanoes. These topics determine the structure of the paper.

## 2  Methods

### 2.1  The dual-beam ScanDOAS instrument and real-time operation

NOVAC is a network of dual-beam ScanDOAS instruments. This is a well-established technique that has been described in detail elsewhere (Johansson et al., 2009; Galle et al., 2010, 2011). There are two types of NOVAC instruments: "Mark I" TS31 are more robust and simpler, designed for routine long-term monitoring, and "Mark-II" instruments, with more sophisticated optics and spectrometer, were developed for more specific scientific observations (Kern, 2009). The results of this study correspond to measurements with Mark-I systems, which comprise more than 95 % of installations (and > 99 % of collected data).

A typical volcano in NOVAC is monitored by two or three ScanDOAS instruments, located within 10 km distance from the main volcanic vent. The objective is to guarantee as complete azimuthal coverage of the volcanic plumes as possible as determined by wind patterns and permitted by logistical constraints. The selection of the sites for installation should also consider aspects of (i) altitude (neither too high to obtain clear atmospheric spectra outside of the plume nor too low to avoid obstacles in the viewing directions of the instrument), (ii) distance from the vent (neither too close, where

turbulence and the optical thickness of the plume may affect the quality of the measurements, nor too far, where atmospheric dispersion and depletion processes take a dominant role making quantification of the source emission difficult), and (iii) orientation of the scanning path (flat or conical, to maximize the probability of intercepting the plume with overlapping scanning paths of several stations, which is used for calculation of plume location by triangulation).

The fore-optics of the dual-beam ScanDOAS instrument consists of a scanning telescopic system with left-handed orientation defining roll (i.e. scanning angle between $-180$ and $+180°$ in steps of 3.6°), pitch (i.e. the conical (60°) or flat (90°) angles of the scanner), and yaw (or azimuth angle, usually oriented towards the volcano). The telescope consists of a single-plane convex quartz lens with a diameter of 25.4 mm and a focal length of 7.5 cm, as well as a Hoya (U330) UV filter that reduces intensity of light with wavelengths longer than 360 nm. The telescope is coupled to one (single-beam) or two (double-beam) quartz optical fibre(s) with a diameter of 600 µm. This combination gives an effective field of view of 8 mrad. The optical fibre is coupled to the entrance slit of the spectrometer, which has a width of 50 µm and height of 1 mm. The spectrometer (SD2000 from Ocean Optics) has a crossed Czerny–Turner configuration with a grating of 2400 lines per millimetre operating in reflection and a UV-enhancement-coated, uncooled, linear charge-couple-device (CCD) detector (ILX511b from Sony) of 2048 14 $\times$ 200 µm effective pixels, as well as a 12 bit analogue-to-digital converter (ADC). The effective spectral range of the spectrometer is $\sim$ 275–480 nm; the spectral resolution (FWHM; full width at half maximum) is $\sim$ 0.5 nm; and the pixel resolution for the combination of grating and slit is $\sim$ 6.5 pixels. The signal-to-noise ($S/N$) ratio at 50 % of saturation is $\sim$ 500 : 1 for an average of 15 spectra taken at typical ($\sim$ 500 ms) exposure time.

Data are transferred via a serial port to the instrument computer. Three versions of the control unit have been developed over the years, with all of them being industrial grade; running on a Linux operating system; and including serial, USB 2, and Ethernet CE10 communication ports. Serial ports are used for communication with the spectrometer and control of the scanner's stepper motor. The USB port can be used for powering the spectrometer, while the Ethernet port is usually used for data transfer to radio modems. Other peripherals include a digital thermometer (for record of internal temperature), a voltmeter (for control of battery voltage), and a GPS antenna (for recording location and time).

The NOVAC instruments are usually powered by an array of 12 V batteries and solar panels; the power consumption at full operation is 6.9–10 W (depending on the model of computer); and communication with the observatories is done by radio modem telemetric networks (usually at 900–930 MHz). A timer is added to interrupt operation of the instrument at night and trigger a reset of the instrument in the morning. Data are collected in situ (a 4 GB CompactFlash CE11 card

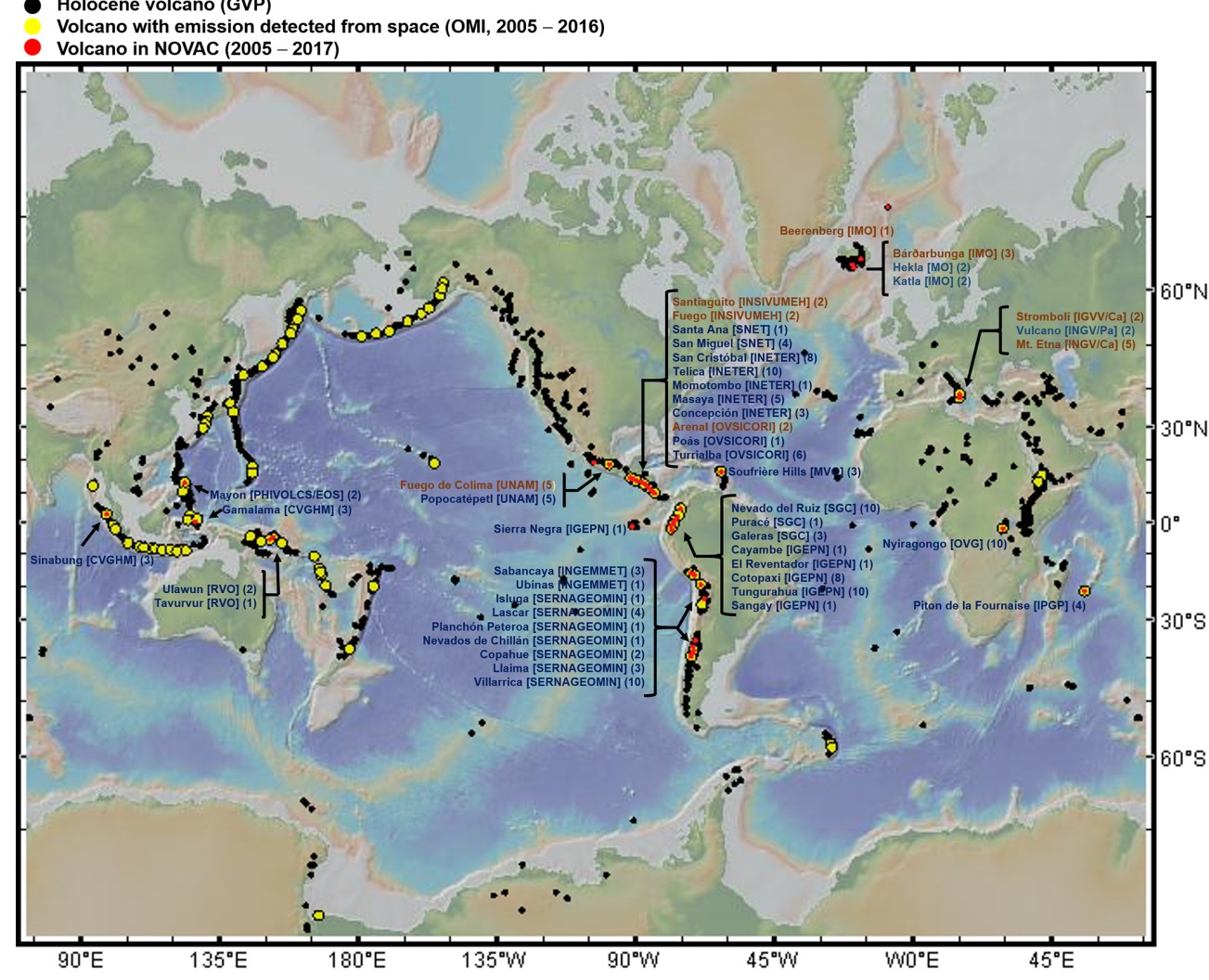

**Figure 1.** Topographic map showing the locations of volcanoes in NOVAC (red circles). The locations of Holocene volcanoes from the Global Volcanism Program (GVP) of the Smithsonian Institution (2013) are shown with black circles. The locations of volcanoes detected by OMI during 2005–2015 (from Carn et al., 2017) are shown with yellow circles. Beside the names of the volcanoes are the acronyms of the volcanological observatories and the number of stations and configurations installed on each volcano over the years. Blue fonts are used to represent volcanoes observed by NOVAC by the time of writing, and orange fonts are for volcanoes observed in the past or where ready-to-deploy infrastructure is in place. For a list of volcanoes, institutions, contact details and links to the database, see Supplement S1 (base map in Mercator projection, from http://www.geomapapp.org, last access: TS30, Ryan et al., 2009). CVGHM: Center for Volcanology and Geological Hazard Mitigation; IGEPN: Instituto Geofísico; IMO: Icelandic Meteorological Office; INETER: Instituto Nicaragüense de Estudios Territoriales; INGEMMET: Instituto Geológico, Minero y Metalúrgico; INGV: Istituto Nazionale di Geofisica e Vulcanologia; INSIVUMEH: Instituto Nacional de Sismología, Vulcanología, Meteorología e Hidrología; IPGP: Institut de Physique du Globe de Paris; IVGG/Ca:; MO:; MVO: Montserrat Volcano Observatory; OVG: Observatoire Volcanologique de Goma; OVSICORI: Observatorio Vulcanológico y Sismológico de Costa Rica; PHIVOLCS: Philippine Institute of Volcanology and Seismology; RVO: Rabaul Volcanological Observatory; SERNAGEOMIN: Servicio Nacional de Geología y Minería; SGC: Servicio Geológico Colombiano; SNET: Servicio Nacional de Estudios Territoriales; UNAM: Universidad Nacional Autónoma de México. CE9

can keep compressed format data for up to ∼ 6 weeks before older data are overwritten) or transmitted for real-time evaluation and display of results with the software NOVACProgram CE12 (Johansson, 2009). Raw and analysed data are archived in a server hosted in Gothenburg and mirrored in Brussels and Heidelberg. This server is accessible to members of the network.

The standard protocol for a measurement of the flux of SO₂ begins with a determination of the exposure time required for an adequate (typically about 65 %) saturation of the spectrometer detector. For this, the scanner is moved to

a 0° scan angle (closest to zenith), and the exposure time is adjusted to a value between 50 and 1000 ms. Next, a preliminary Fraunhofer reference spectrum is measured at a 0° scan angle; then a dark spectrum is recorded at a 180° scan angle (closest to the obstructed-view nadir), followed by a total of 51 measurements of skylight from scanning angles −90 to 90° at steps of 3.6°. Each measured spectrum consists of 15 co-added spectra to increase $S/N$. A full scan is collected every 1–15 min, depending on illumination conditions. Data are spectrally analysed using the DOAS method (Platt and Stutz, 2008) and evaluated in the spectral range 310.6–324.6 nm. Before the analysis, generic corrections for dark current, electronic offset, and wavelength shift (based on absorption features of SO$_2$) are applied. The spectral analysis model includes absorption spectra of SO$_2$ at 293 K and 1000 mbar (Bogumil et al., 2003) and O$_3$ at 223 K and 100 mbar (Voigt et al., 2001), as well as a ring-effect pseudo-absorber synthesized from the Fraunhofer spectrum using the software DOASIS (DOAS Intelligent System; Kraus, 2006) and a fifth-degree polynomial to account for broadband extinction. Molecular absorption cross-section spectra are retrieved from the MPI-Mainz UV/VIS Spectral Atlas of Gaseous Molecules of Atmospheric Interest CE13 (Max Planck Institute; ultraviolet–visible spectrum; Keller-Rudek et al., 2013). For the DOAS analysis, a convolution is applied to the high-resolution spectra with the instrumental function of each instrument, which is approximated from the 302.15 nm emission line of a low-pressure Hg lamp measured at room temperature before installation. All calibration data are archived in the data server.

Along with instrument endurance, data acquisition and analysis have also been designed to guarantee compensation of measurement errors and traceability of measurement conditions. The compensation is achieved by acquiring reference and dark spectra on each scan, which permits an efficient cancellation of instrumental imperfections, which may change over time because the measurements are taken within minutes of each other. The traceability is ensured by logging all measurement parameters, which allows for rigorous scrutiny of the quality of the measurements and offers the possibility of applying more advanced algorithms in the future that make use of this auxiliary information (e.g. instrument line shape models that account for the effect of temperature).

The NOVAC instruments have proven to be remarkably robust, particularly given the harsh conditions they are regularly exposed to. Instruments are often installed at high elevation, exposed to large temperature and humidity variations, and experience ash fall or even exposure to highly acidic volcanic gases. The simple design of the instruments and the separation of the optical scanner from the rest of the instrumentation are key to their robustness. This, combined with the strong sense of community within the NOVAC consortium, has led to the growing number of scanners installed at active volcanoes around the world.

## 2.2 Batch processing with the NOVAC Post Processing Program

As mentioned above, data are transmitted to the observatories, where they are analysed and archived in real time using the NOVACProgram. This evaluation uses meteorological information (wind speed and direction) provided by each operator; it may thus vary enormously in quality among different observatories. Additionally, the combination of nearly simultaneous measurements from intercepting scanning paths of two instruments is used to calculate plume height and direction in real time. Also plume speed can be derived from measurements of a single instrument by the dual-beam cross-correlation method when certain conditions are fulfilled regarding direction, strength, and stability of the plume (Johansson et al., 2009b TS32; Galle et al., 2010).

In order to adopt a standardized methodology for the evaluation of data collected by each station, a programme called the NOVAC Post Processing Program (NovacPPP) was developed by Johansson (2009). This programme retrieves all scan measurements collected at a specified volcano within a given period and proceeds to evaluate them selecting the best information available for each variable. For instance, measurements of plume speed, direction, and height are prioritized over information obtained from a meteorological model or from common assumptions (e.g. plume height equal to difference in altitude between volcano summit and scanner).

In this work we used wind speed from the ERA-Interim re-analysis database of ECMWF, which is based on the Integrated Forecasting System (IFS) Cycle 31r2 4D-Var (variational) assimilation system, using a TESSEL (Tiled ECMWF Scheme for Surface Exchanges over Land) land-surface model. This database, with a coverage period since 1979 until 2019, has an assimilation period of 12 h, while the spatial resolution is 79 km (TL255) in the horizontal and has 60 vertical levels from sea level pressure up to 10 hPa, with a typical difference equivalent to ∼ 200 m (Dee et al., 2011). For each volcano, horizontal wind vectors, relative humidity, and cloud cover are retrieved every 6 h on a horizontal grid of 0.125 × 0.125° (13.9 × 13.9 km for mean Earth radius) surrounding the location of the main volcanic vent. It is then further interpolated to the vent location and the time of each scan.

The programme also applies a correction for spectral shift (i.e. the possible change in the pixel-to-wavelength mapping of the spectrometer during operation), based on correlation of the position of the Fraunhofer lines in the measured spectrum with those of a high-resolution solar spectrum (Chance and Kurucz, 2010) adjusted to the resolution of the spectrometer. Other than these changes, the evaluation follows the same routines as the standard settings of the NOVACProgram, specified above.

## 2.3   Uncertainty of SO$_2$ mass flow rate measurements

It is difficult to assign a "typical" uncertainty to measurements of flux with NOVAC instruments because the flux calculation depends on different variables and assumptions, which are subject to a wide range of conditions (meteorology, distance to plume, content of aerosols, amount of absorber, etc.). Detailed analysis performed by Arellano (2014) shows that the range of uncertainty can be as low as 20 %–30 % or as large as > 100 %. Categories of uncertainty include model, measurement, and parameter uncertainties. Model uncertainty refers to the plausibility that a certain measurement scenario is realized in practice. For example, the assumption that transmittance can be calculated from simple application of the Beer–Lambert–Bouguer law may not hold due to radiative transfer effects (Millán, 1980; Mori et al., 2006; Kern et al., 2010a), or the model adopted for the geometrical shape of the plume may not be adequate. Measurement uncertainty could be induced, for example, by inaccurate determination of the viewing direction of the scanner or variations in the spectrometer response caused by changing environmental conditions. Parameter uncertainty could e.g. be caused by inaccuracy of a laboratory absorption cross section or the uncertainty in plume speed data derived from a mesoscale meteorological model.

If we split the analysis into the variables involved in the calculation of a single flux measurement, the sources of uncertainty include the uncertainty in the derivation of the column densities, plume speed, plume height, plume direction, orientation or scanning angles, and radiative transfer. If the intention is to quantify the source emission strength from the measurements of the plume mass flow rate, the possible depletion/production of SO$_2$ downwind the vent, understood as the sum of all processes that reduce/increase the measured amount of SO$_2$, should be further considered. Measurement and parameter uncertainties can to a large extent be derived from the actual observations and the literature. Analysis presented in Arellano (2014) indicates that ScanDOAS measurements have asymmetric distribution of uncertainty, showing typically high left skewness; i.e. the mean value of the distribution is most likely an underestimation of the true flux. In the results presented here, we compute statistics of daily flux only for measurements considered to have "good quality", based on several criteria, specified below. By adopting these criteria, we consider that a reasonable minimum estimate of fractional uncertainty lies between −30 % to 10 %; i.e. the reported values for individual flux measurements correspond to the average value, while the span of the uncertainty has 1$\sigma$ limits of confidence between 70 % and 110 % of the average value. The reduction in the number of valid results is usually large (40 %–60 % of the total number of measurements, depending on the site). As the intention of this paper is to improve the statistics of measurements of SO$_2$ emission, we think that negotiating more quality for less quantity is justi-

fied. For details of the available raw data, see Fig. S1 in the Supplement.

## 2.4   Criteria for data selection

Each measurement considered for the statistics and analyses presented in this paper has been validated according to a few quality criteria. As a prerequisite, valid spectra have position, time, total duration ($\leq$ 15 min), and observation geometry all within normal ranges. We tracked the history of installation of each station, determining the locations, orientations, and scanning geometries of each spectrometer over the years, and checked them for accuracy. Then, each spectrum should have adequate intensity, neither saturated nor over-attenuated ($\leq$ 10 % of saturation) in the region of evaluation ($\sim$ 310–325 nm). The DOAS fit threshold for retrieval of SO$_2$ corresponds to a chi-square ($\chi^2$) value of $9 \times 10^{-3}$. For a plume scan to be included in the analysis, we required it to have a "plume completeness", calculated according to the algorithm described in Johansson (2009), of at least 0.8. The absolute value of the scan angle with respect to zenith of the column-density-weighted centre of mass of the plume should not be larger than 75°, and the calculated plume geometry should be reasonable (e.g. measurements which retrieved a distance to the plume of larger than 10 km are not considered for further analysis). From the set of valid flux measurements in a day, time-averaged statistics are computed if at least five measurements passed the quality checks. Figure 2 shows a flow chart of the steps followed in the evaluation of data.

# 3   Results

## 3.1   SO$_2$ emission rate

In this study we report daily statistics of SO$_2$ emission. These are derived from minute-scale scan measurements, but we regard the daily emission as more representative of volcanic degassing because the sub-daily values are subject to large variability introduced by meteorological effects and tidal influences and other reasons (e.g. Bredemeyer and Hansteen, 2014; Dinger et al., 2018). We report the daily average SO$_2$ emission rate, standard deviation, different quantiles, and number of measurements in each day, as well as similar statistics for plume location, velocity, and cloud cover. Figure 3a–h show the time series of daily SO$_2$ flux between 1 March 2005 and 31 January 2017 for 32 volcanoes in NOVAC which produced a reasonable amount of valid data. Figure 4 shows the mean emission and 25 %–75 % quantiles calculated from measured fluxes for all volcanoes during the same period. Results in numerical format are presented in Supplement S2. An important exception in this compilation is Bárðarbunga volcano in Iceland; its Holuhraun eruption in 2014–2015 was monitored in detail, but the analysis of its data required special handling not apt for the procedure

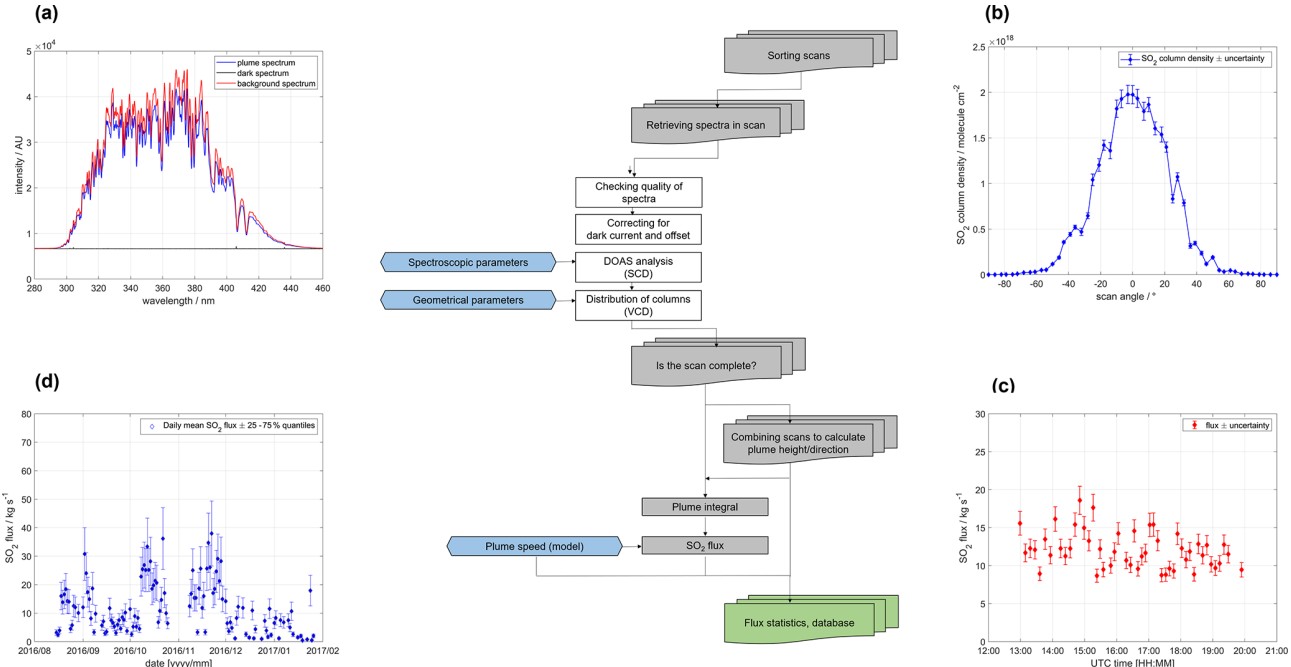

**Figure 2.** Schematics of the algorithm used to derive time-averaged emission of volcanoes in the NOVAC database. **(a)** Scattered sunlight spectra (shown in the figure as uncalibrated spectral radiance in arbitrary units; AUs) are checked for quality and combined to correct instrumental effects and to derive the differential slant column density (SCD) of $SO_2$ through the non-linear DOAS method. **(b)** A collection of column densities in the scan is used to determine the baseline column density and the angular position of the centre of mass of the plume, to convert the slant to the vertical column density (VCD), and to estimate the completeness of the scanned plume. **(c)** Pairs of scans taken close in time by different instruments are used to derive plume altitude and direction. This is combined with plume speed using a meteorological model to derive the flux, including uncertainty. Individual flux measurements are chosen considering uncertainty, completeness, and other criteria. **(d)** If at least five valid measurements exist on a given day, statistics of daily emission are computed and reported in the NOVAC database. The background colour of the boxes indicates processing at the spectral level (white), scan level (grey), flux level (green), and external parameters (blue).

described here due to extreme measurement conditions and enormous amounts of gas (Pfeffer et al., 2018).

## 3.2 Long-term emission budgets and comparison with satellite-based data

The analysis of long-term data from automatic instrumental networks of this type presents a challenge for the extrapolation of (often irregular) sets of measurements in producing an estimation of time-averaged emissions. This challenge has to do with distinguishing periods of null observations, in the sense described above (i.e. when less than five measurements of good quality were obtained within a day), or those which are caused by instrumental (e.g. when no measurements were acquired) or observational (e.g. winds drifting the plume beyond zone of observation) causes, from periods of legitimate low emission (i.e. absence of a plume). To account for these periods, we need additional information about the level of activity, visual observations, photographic records, etc., which is not always available.

To deal with this problem, we have adopted the following strategy: if there are no statistics of emission for a given day

and there were either no scanning measurements conducted or the mean plume direction (obtained from the meteorological model) lies outside the 5 %–95 % range of historical plume directions observed by the instruments, then no inference can be made about the actual emission on that day, and the value is simply interpolated linearly between the nearest data points with valid observations. On the other hand, if measurements were done and the modelled wind data indicate that the plume should have been observed by the instrumental network, we attribute the lack of data for that day to low volcanic emission. The actual value of this low emission is chosen as the 5 % quantile of valid historical observations. This value is chosen arbitrarily to represent an effective detection limit, noticing that the flux depends not only on the actual gas column density detection limit but also on the size and speed of the plume.

By filling in data in this way we can obtain a more regular and accurate representation of the actual emission for prolonged periods of time and calculate the corresponding statistics. The results of this procedure are shown in Fig. 5a–c, where we present only the time series for volcanoes which have corresponding observations by OMI in the same period

https://doi.org/10.5194/essd-13-1-2021

Earth Syst. Sci. Data, 13, 1–27, 2021

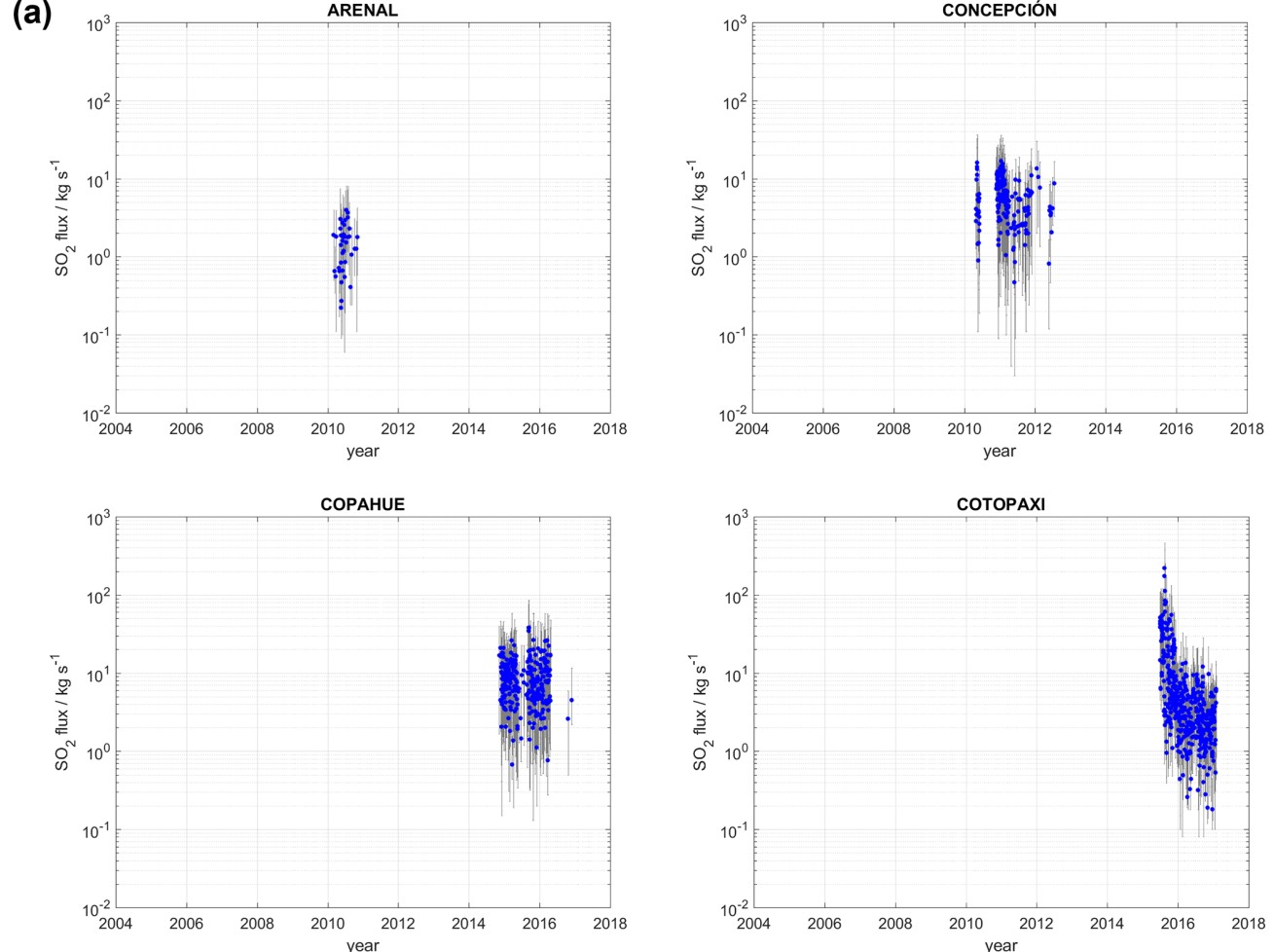

**Figure 3.**

(12 out of 32) as reported by Carn et al. (2017). Notice also the time series of "only observed" data along with the corresponding time series of emission from the OMI sensor. Results in numerical form are presented in Supplement S1.

## 3.3 NOVAC emission data repository

The results presented here are made public through a data repository hosted on the website https://novac.chalmers.se/ (last access: TS33). The site shows a map with the location of the volcanoes for which valid data have been produced. The dataset produced according to the methodology described here is labelled Version 1, and updates (temporal increments) and upgrades (different versions of data produced with improved methodology) are planned in the future. The dataset shows a summary of available raw data (i.e. scans) collected by the instruments, along with a summary of valid fluxes derived from those measurements.

After selection of a volcano, a dedicated window presents a map with the setup of monitoring instruments, including coordinates and measurement parameters, a link to generic information about the volcano hosted on the Smithsonian Institution's Global Volcanism Program website (https://volcano.si.edu/, last access: TS34), information on the responsible observatory and contact details, and the time series of daily mean $SO_2$ emissions with associated statistics. The plots are easy to explore through different scaling and textual information. From each volcano page, data can be downloaded, after registering basic contact information and accepting the data use agreement, which states (e.g. for the case of Popocatépetl volcano) the following:

CE14 Large efforts have been made by the volcano observatories and institutions responsible for data collection and evaluation. Thus, data presented here can be used on the condition that these organizations and people are given proper credit for their work, following normal practice in scientific communication:

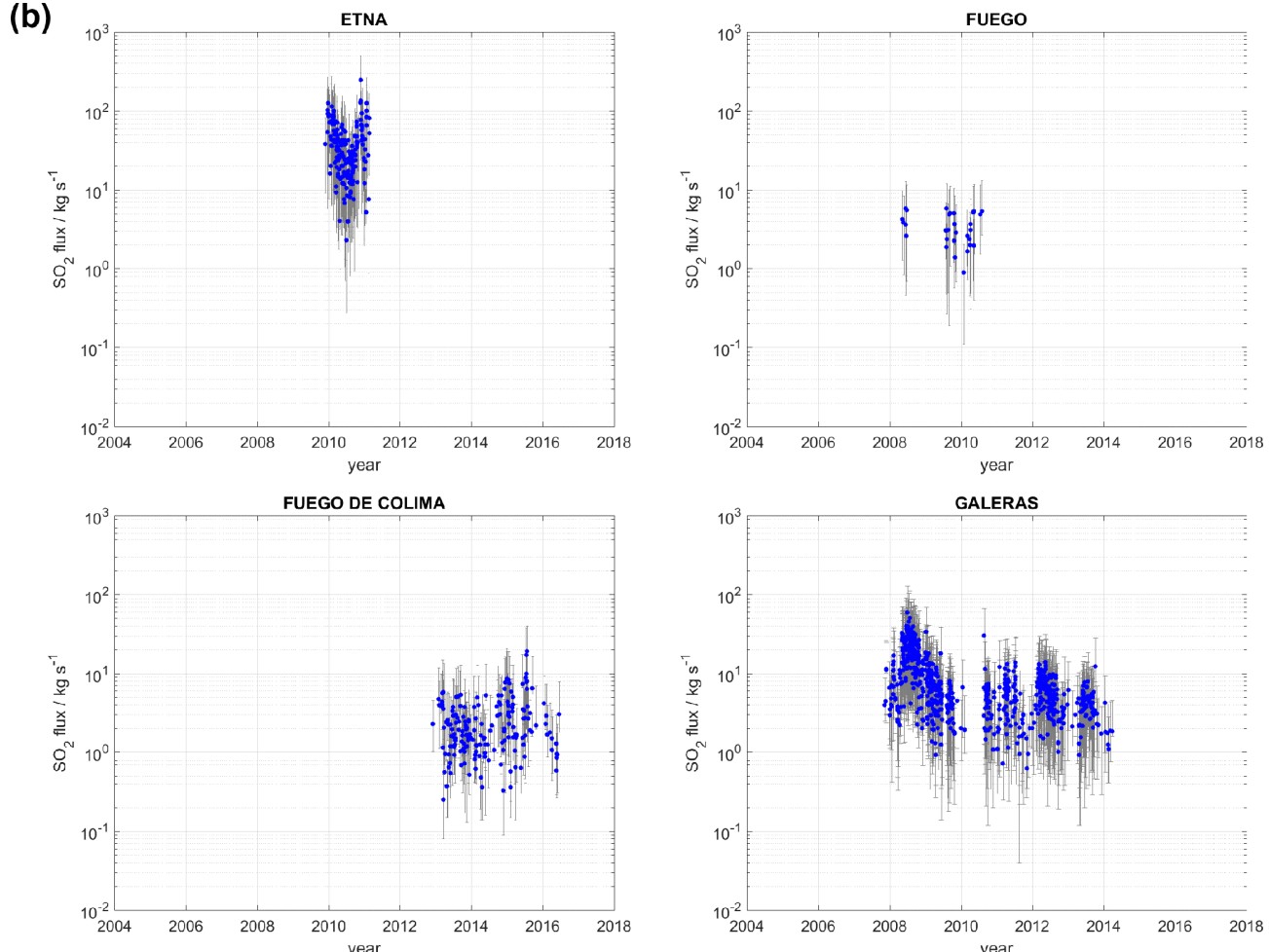

**Figure 3.**

(1) If data from this repository contributes an important part of the work, co-authorship should be offered to the listed contributors and the data-set should be cited.

(2) If data from this repository contributes only a small, but still important part of the work, the data-set should be cited.

To cite this data-set include this information:

*******************************************

Delgado, H., Arellano, S., Rivera, C., Fickel, M., Álvarez, J., Galle, B., SO₂ flux of -POPOCATEPETL- volcano, from the NO-VAC data-base; 2020; [Data set]; v.001; doi:10.17196/novac.popocatepetl.001

*******************************************

Additional data, data with higher time resolution and raw data may be made available upon request to the respective contacts, listed below.

Notice that each dataset is assigned a registered and permanent digital object identifier (DOI).

The data files were prepared following the guidelines of the Generic Earth Observation Metadata Standard (GEOMS) (Retscher et al., 2011), which are generic metadata guidelines on atmospheric and oceanographic datasets adopted for global initiatives, such as the Network for Detection of Atmospheric Composition Change (NDACC). The GEOMS standard requires a file format such as netCDF. This data format can be explored using openly available tools such as Panoply (https://www.giss.nasa.gov/tools/panoply/, last access: TS35). For users not familiarized with the netCDF format, a text format file, easily accessible through standard workbook or text editor applications and containing the same information that the netCDF file, is also available. An example of such a file is presented in Supplement S3, and a list of all files is listed in Supplement S1.

The GEOMS standard requires data and metadata to be included in the same file. In the case of NOVAC, the metadata,

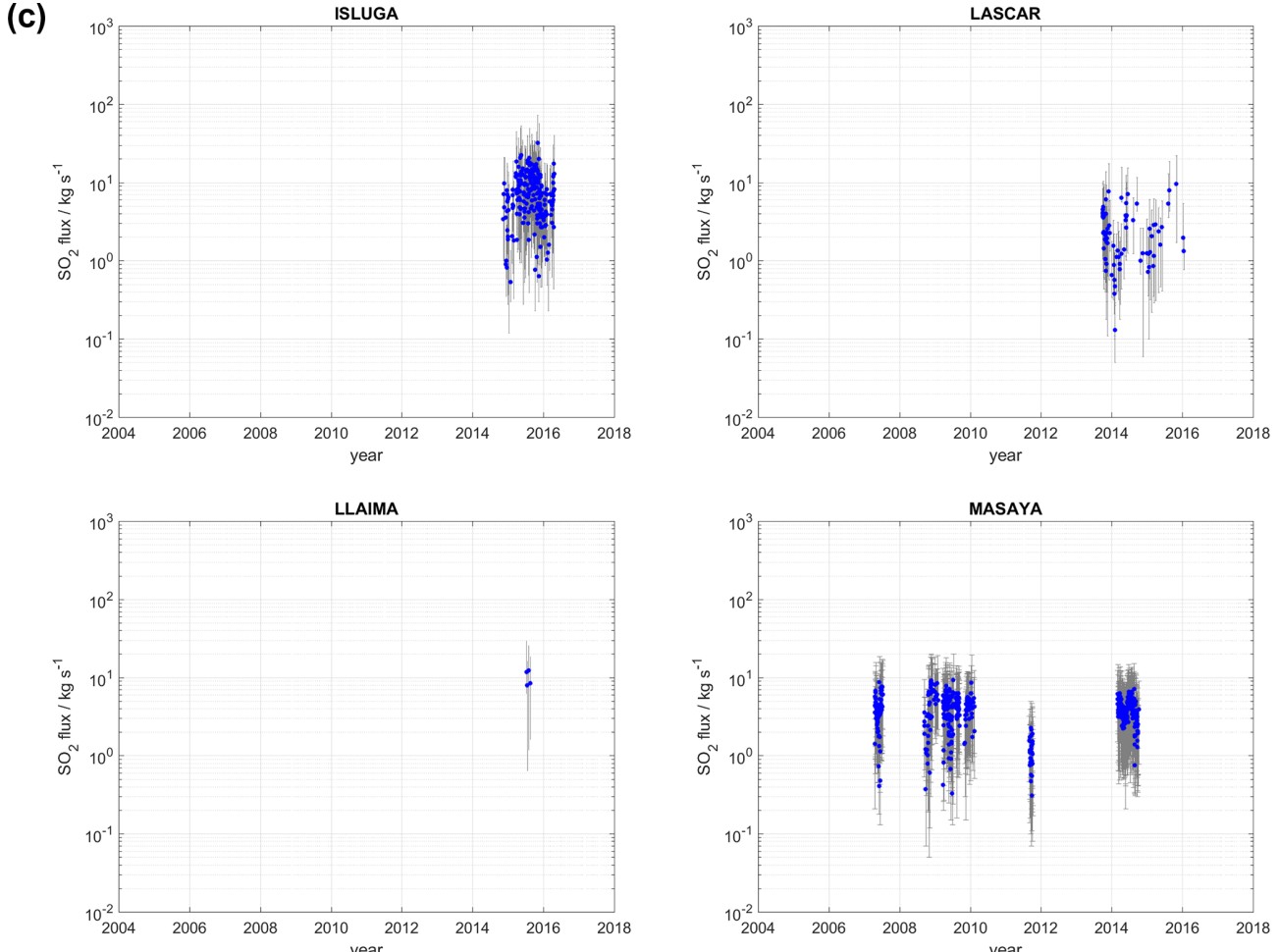

**Figure 3.**

required for the description and interpretation of the data, include the following:

- general information about the dataset

- data use agreement

- data set description (site, measurement quantities, processing period, processing level, data version, DOI, accompanying file, and date of file production)

- contact information

- reference articles

- instrument(s) description (instrument type, spectrometer specification, fore-optics specifications, control unit specifications, instrument ID(s), site name(s), site coordinates, site measurement parameters, and instrument serial numbers)

- measurement description

- algorithm description (slant column densities, vertical column densities, SO$_2$ flux and plume parameters, and statistics)

- expected uncertainty of measurement

- description of appended results.                    20

The data include the following:

- date according to universal time (UT)

- daily mean, standard deviation, quartiles, and number of valid SO$_2$ flux measurements

- daily mean and standard deviation of plume speed      25

- daily mean and standard deviation of plume direction

- daily mean and standard deviation of plume height

- daily mean and standard deviation of plume distance to instruments and width

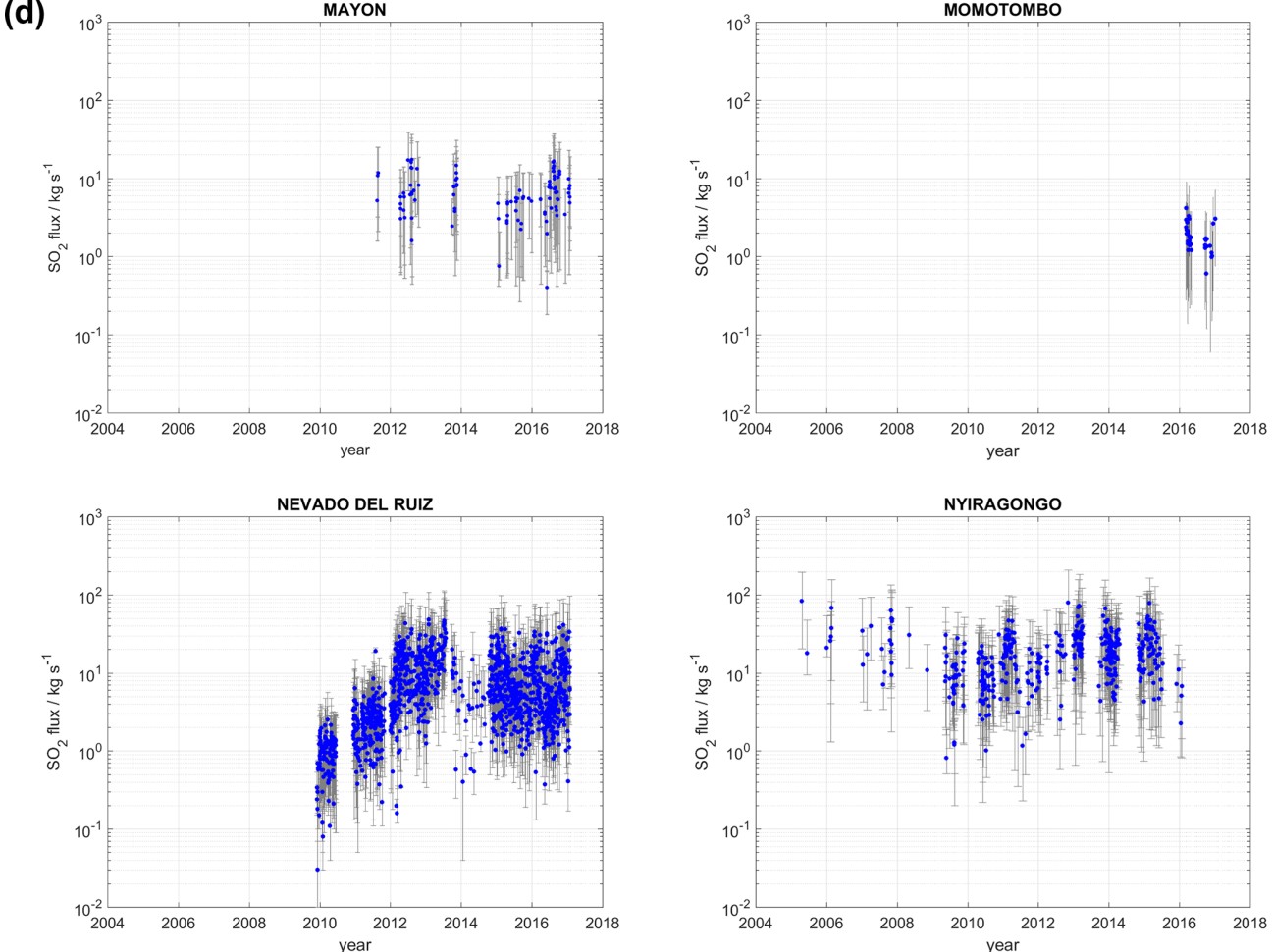

**Figure 3.**

– daily mean and standard deviation of cloud cover (from re-analysis meteorological model).

Additional pages in the data repository provide details about the database, the data use agreement, technical details of the instrument, description of algorithms used for available data versions, contact information, and acknowledgements.

The NOVAC data repository will be linked to other thematic databases such as the Database of Volcanic Unrest (WOVOdat) of the World Organization of Volcano Observatories, the database of the Global Volcanism Program of the Smithsonian Institution, the EarthChem data repository, the Global Emission InitiAtive (GEIA), the database of the Emissions of atmospheric Compounds and Compilation of Ancillary Data (ECCAD), and the database of the EU Copernicus Atmospheric Monitoring Service (CAMS).

## 4 Discussion

### 4.1 Comparison of emission from different volcanoes

Figures 3a–h and 4 summarize the statistical information about the time series of emission for 32 volcanoes in NOVAC during 2005–2017. The plots show the daily and annual means and 25 %–75 % quantiles of daily SO$_2$ emission to represent variability. We highlight three main characteristics from these results: (i) the relatively large range of variation of emissions, spanning typically up to 3 orders of magnitude in variability, for the same volcano at different times (Fig. 3a–h); (ii) the skewed nature of the distributions, with a dominance of low emission values (i.e. more frequent low emission rate values and a few large emission values that account for a considerable fraction of the total emission); and (iii) the large difference between the characteristic emission of different volcanoes (Fig. 4).

With respect to the intra-variability (for a particular volcano), we consider this to be one of the most important findings of long-term monitoring. As mentioned in the Introduc-

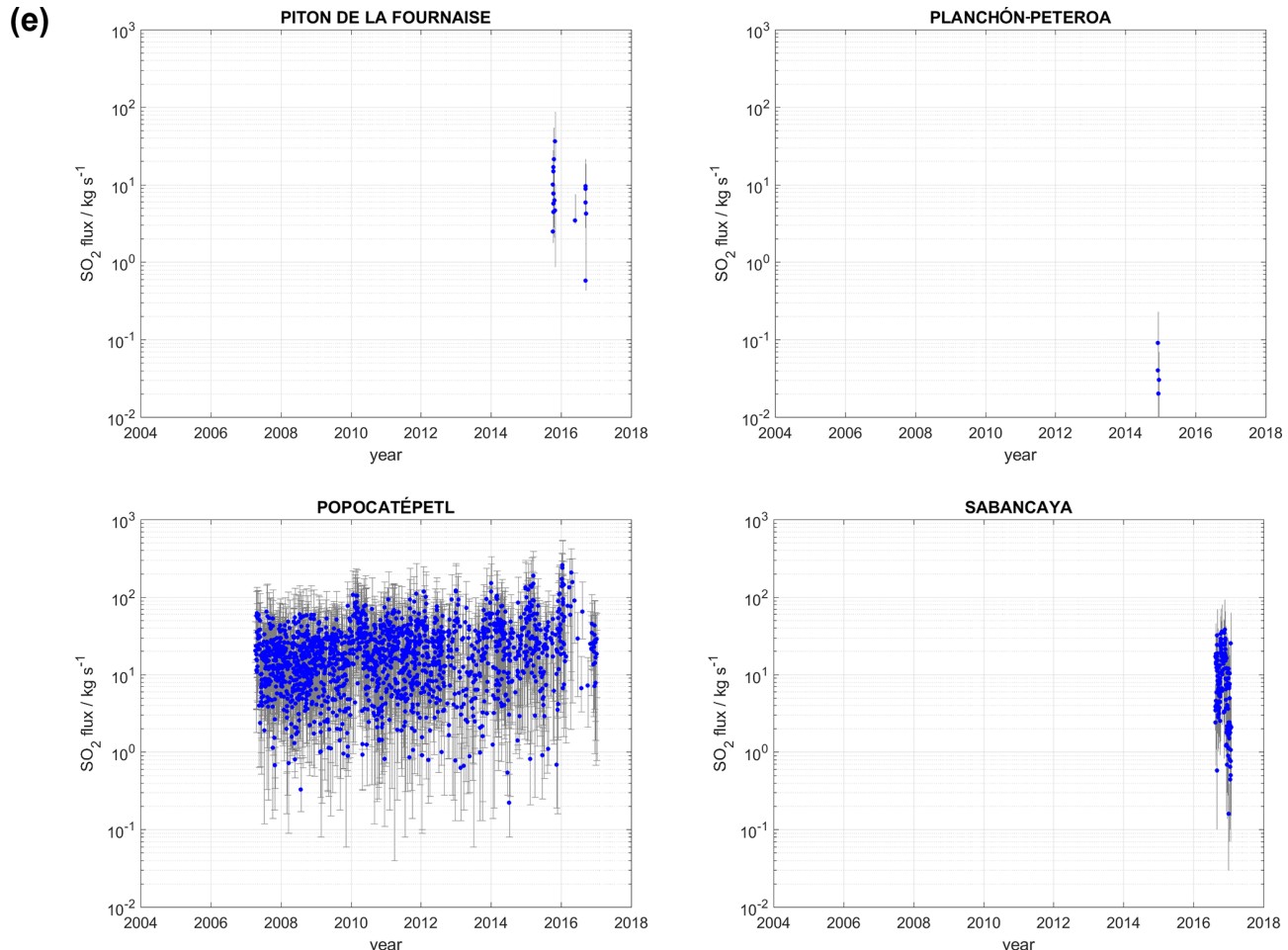

**Figure 3.**

tion, the production of high-sampling-rate, long-term measurements is relatively recent. Most compilations of measurements in the past include campaign-based estimates of gas emissions, typically during periods of enhanced activity, when a plume was visible and during short periods of time. The skewed, large-range distributions of emission seem to be a general feature of degassing volcanoes and merit more attention. Our analysis, using only measured, i.e. not "filled-in", data, indicates that the ratio of the first quartile to the mean of daily SO₂ fluxes reaches $43 \pm 14\%$ ($\pm 1\sigma$), which means that the distribution of daily emission is dominated by low values. An important implication of this finding is that the low-emission spectrum of the distribution, which has usually not been measured in the past, contributes a significant amount of the total emission and should therefore be better characterized. Another is that short-term measurements may be skewed and could therefore not be representative of the long-term emission of a volcano.

Regarding the inter-variability (among different volcanoes), the observation of a large variance between sources is not new. Indeed, it has been speculated and partially shown

by several authors (e.g. Brantley and Koepenick, 1995; Andres and Kasgnoc, 1998; Mori et al., 2013; Carn et al., 2017) that the partition between sources of volcanic degassing, particularly quiescent degassing, seems to follow either a log-normal or a power-law distribution. These distributions may seem similar, but choosing one over the other results in significant differences in estimating the global volcanic flux. The relative importance of low vs. high emitters is also different for log-normal or power-law distributions. Evidently, with 32 volcanoes, out of perhaps 90–150 degassing volcanoes, it is not possible to verify these speculations with certainty. In any case, our measurements provide bounds for the contribution of weak emission sources, which have escaped observation by satellites during the same period.

## 4.2    Ground-based vs. space-based observations

The recent compilation of global volcanic degassing from satellite-based measurements of OMI (Carn et al., 2017) offers an excellent opportunity for comparison with the measurements obtained from the ground with NOVAC instru-

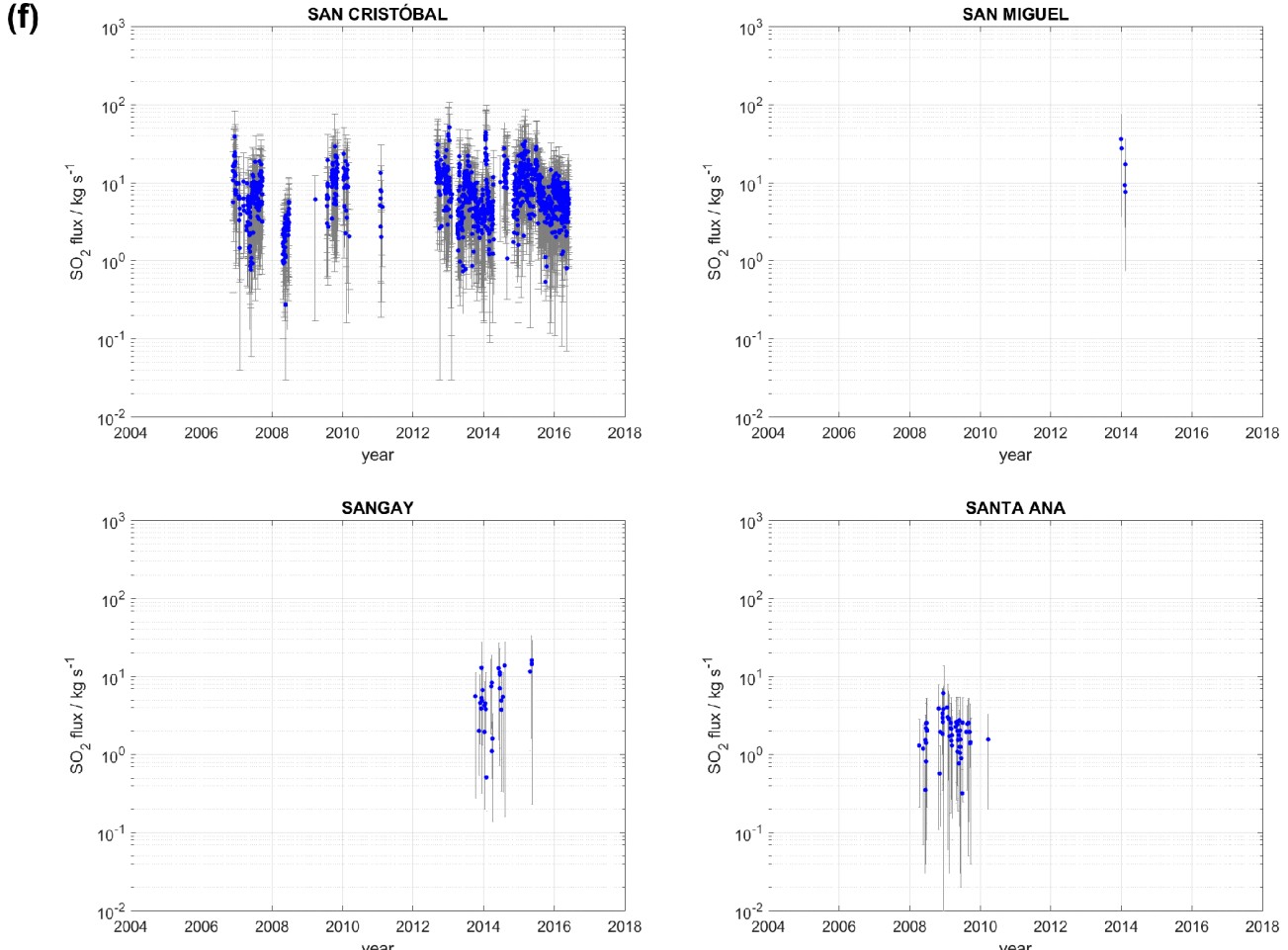

**Figure 3.**

ments. First, both methods have operated for about the same period (since 2005); second, both sets of measurements are analysed independently in a consistent manner; and, third, the two datasets are focused on passive degassing. The quantification of SO$_2$ flux from OMI observations was achieved by stacking of wind-rotated images over the course of a year, discarding both pixels contaminated by clouds and pixels with elevated column densities resulting from explosive eruptions. The co-added annual image is then fitted to a Gaussian distribution, following Fioletov et al. (2016), and the goodness of this fit is expressed as an "uncertainty", but the actual uncertainty in the reported emission is not quantified but assumed in the order of 50 % (Fioletov et al., 2016; Carn et al., 2017).

As mentioned above, it is necessary to fill in the measured SO$_2$ emissions at each volcano during times when degassing was not detected but the instruments would have picked it up had it been occurring. The original, "un-filled" time series and the time series "filled" with low emission values are pre-

sented in Fig. 5a–c, along with the corresponding time series for OMI.

This comparison shows a general agreement in the temporal trends of annual emission for ground- and space-based methods but with differences in magnitude, which in some cases are considerable. Only 12 out of 32 volcanoes from NOVAC have corresponding detections from OMI. This is not surprising, as all volcanoes not observed by OMI are weak sources of emission, confined to the lower atmosphere and in some cases located in areas of persistent cloud cover. Consequently, our dataset provides new data for several volcanoes, such as Sangay, Cotopaxi, Planchón-Peteroa, and Sinabung, and the largest dataset for all other volcanoes ever published.

Figure 5a–c, as well as Supplement S1, also show that the difference between ground- and space-based observations is reduced by the method of filling in low emission values in the patchy time series in NOVAC. Notwithstanding this better convergence, the differences are, in general, biased towards higher emission observed from satellites. There are

**(g)**

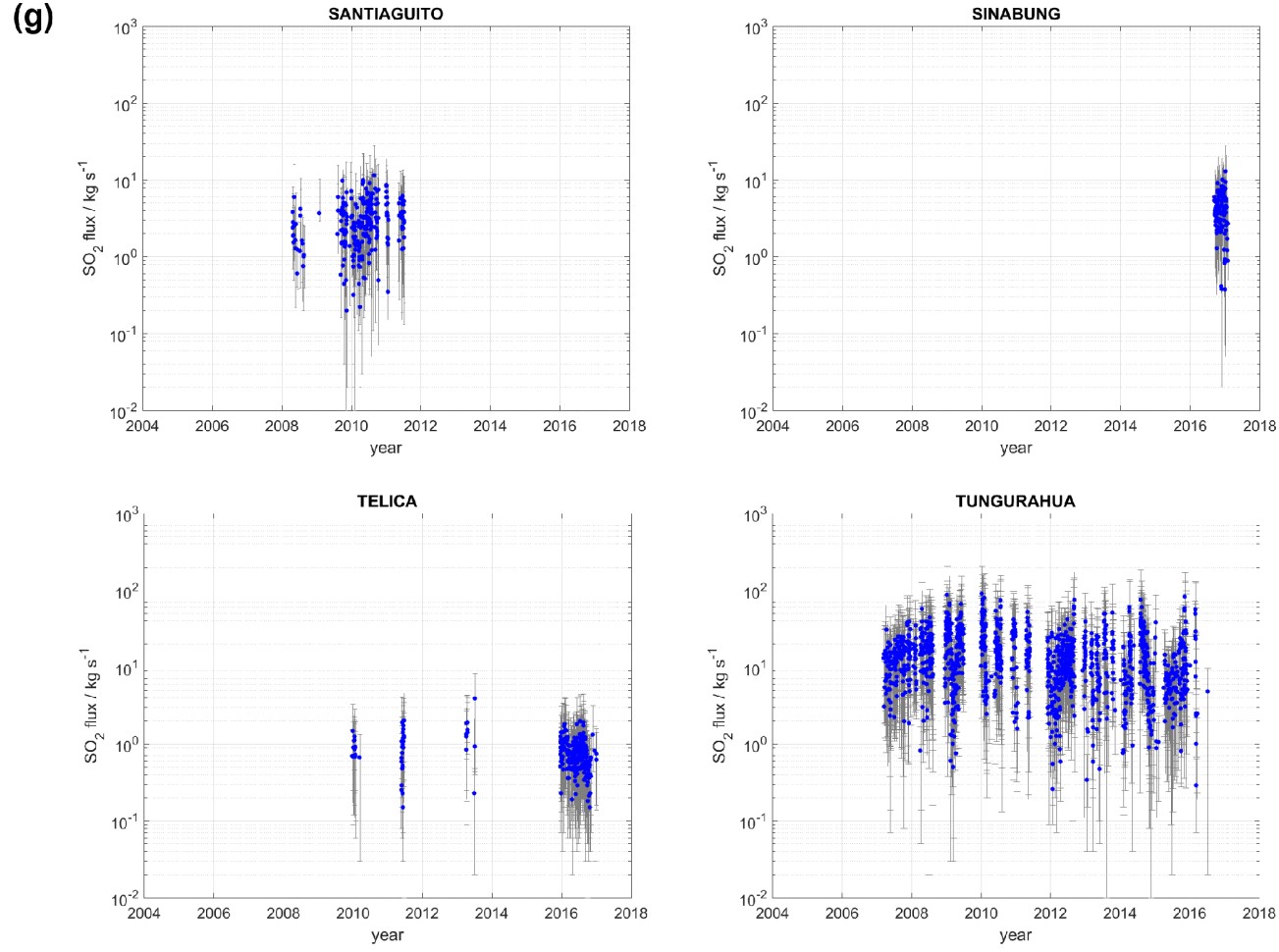

**Figure 3.**

many possible reasons for this; for example, the selection of data for OMI may tend to pick images from higher plumes that reach altitudes above low-level clouds, which may be the result of more explosive activity. Another reason could be that the data selection for NOVAC favours plumes with clear boundaries, and in some instances, high-gas-content plumes may be too wide and thus completely overcast the instruments, and, as a consequence, these are filtered out by the strict quality control filters applied to the dataset during our analysis.

However, the larger differences are caused by obvious reasons: for example, in the case of two nearby volcanoes, such as Nyiragongo–Nyamuragira CE15 (with a footprint of $13 \times 24\,\text{km}^2$), OMI cannot separate completely the contributions of each source, so they are reported as a complex. In this respect, NOVAC can aid in discriminating between these sources, since the stations are deployed with a focus on Nyiragongo and the finer time resolution allows disentangling contributions, especially during periods of heightened activity at any of them. Other reasons for discrepancy are to be found in the different periods covered by the instru-

ments, i.e. only daytime measurements for NOVAC, whereas OMI could in principle detect the emission occurring while overpassing at 13:30 LT in addition to remaining gas that was emitted emissions during the previous hours, potentially even during night. Other factors are the relatively large measurement uncertainties of both methods and different radiative transfer effects depending on altitude of surrounding plumes. A more in-depth study of these discrepancies is highly needed.

Finally, the method proposed here to account for days with null observations improves considerably the comparison with OMI in general. This is more obvious for volcanoes with constant emissions and good instrumental coverage, resulting in more valid measurements (represented by the size of the circles in Fig. 5a–c), which give us confidence in the validity of this approach.

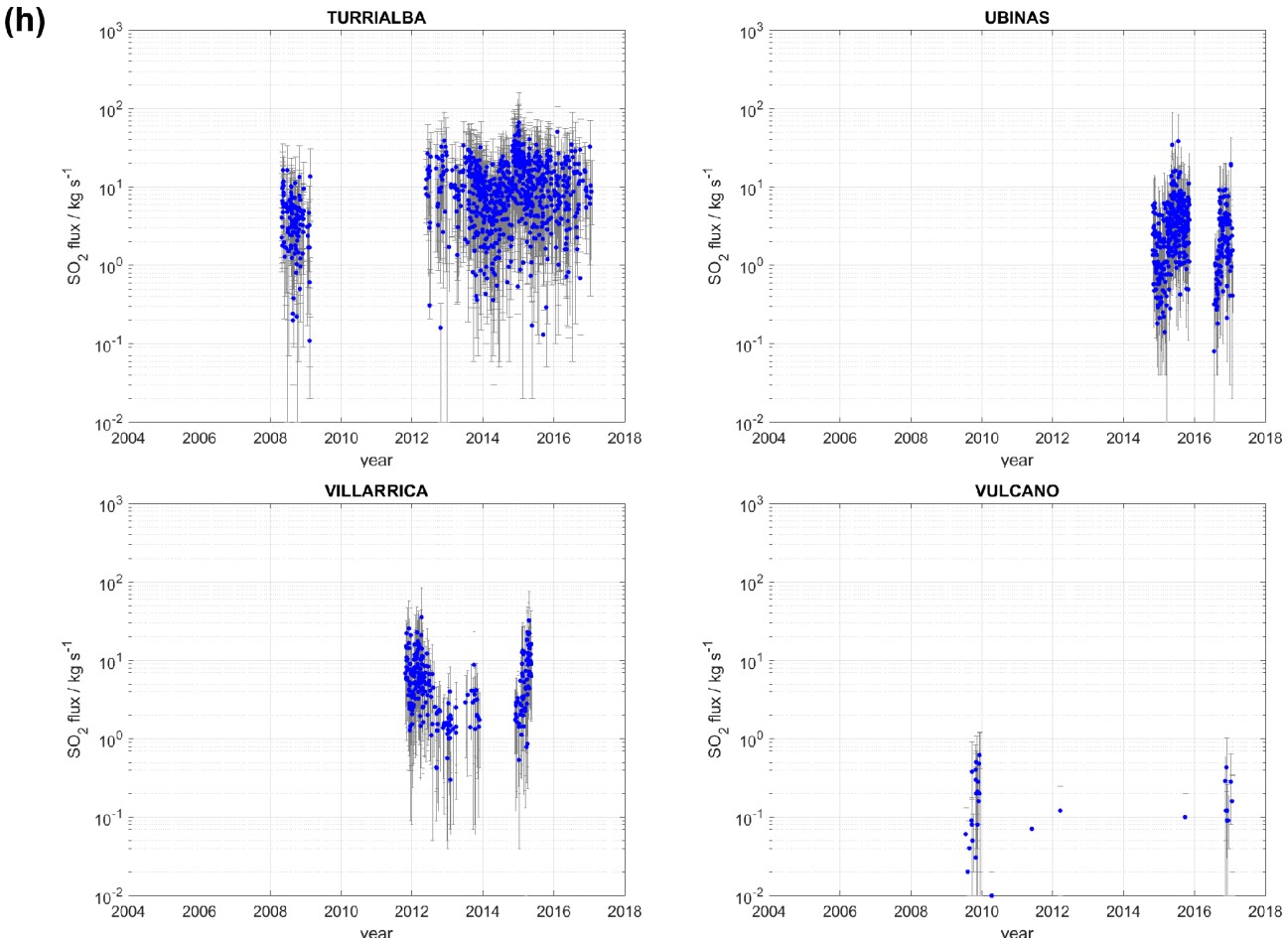

**Figure 3.** Time series of daily $SO_2$ emission from 32 volcanoes in NOVAC from 1 March 2005 to 31 January 2017. Blue dots depict the daily mean emission rate, and grey lines indicate the 25 % and 75 % quantiles of daily flux. References for these datasets are given in Table 1. **(a)** Data from Arenal, Concepción, Copahue, and Cotopaxi. **(b)** Data from Etna, Fuego, Fuego de Colima, and Galeras. **(c)** Data from Isluga, Lascar, Llaima, and Masaya. **(d)** Data from Mayon, Momotombo, Nevado del Ruiz, and Nyiragongo. **(e)** Data from Piton de la Fournaise, Planchón-Peteroa, Popocatépetl, and Sabancaya. **(f)** Data from San Cristóbal, San Miguel, Sangay, and Santa Ana. **(g)** Data from Santiaguito, Sinabung, Telica, and Tungurahua. **(h)** Data from Turrialba, Ubinas, Villarrica, and Vulcano. All data are accessible through the links in Table 1.

### 4.3 The NOVAC inventory and past compilations of emission

It is interesting to compare the emission statistics obtained from the NOVAC data with past compilations of emissions presented in other studies. We refer in particular to Andres and Kasgnoc (1998), who report the volcanic input during 1970–1997 to the Global Emission Inventory Activity (GEIA) database.

The results of a one-to-one comparison between the emissions reported for quiescent degassing volcanoes in GEIA and NOVAC are presented in Fig. 6. There are a few volcanoes reported in GEIA which are not part of NOVAC yet; conversely, some volcanoes were monitored in NOVAC that were not active and thus not considered during the period reported in GEIA. A comparison can only be done for the 16 volcanoes present in both datasets. Undoubtedly, the reported values are not expected to coincide, considering that the measurements were not obtained during the same periods, and, as revealed by the NOVAC results, volcanic gas emission is by no means a stationary process over time. However, it is important to highlight that the recent measurements from NOVAC provide a characteristic range of variation for the volcanic sources, which in most cases, but not all, accommodate the results of past, punctuated observations. But we notice also that, except for Momotombo, San Cristóbal, and Telica, the mean emissions reported in GEIA lie on the upper end or higher than those reported here. We speculate that such systematic difference may be due to biased sampling during periods of high emission that was the basis for most of the Andres and Kasgnoc (1998) compilation. Long-term

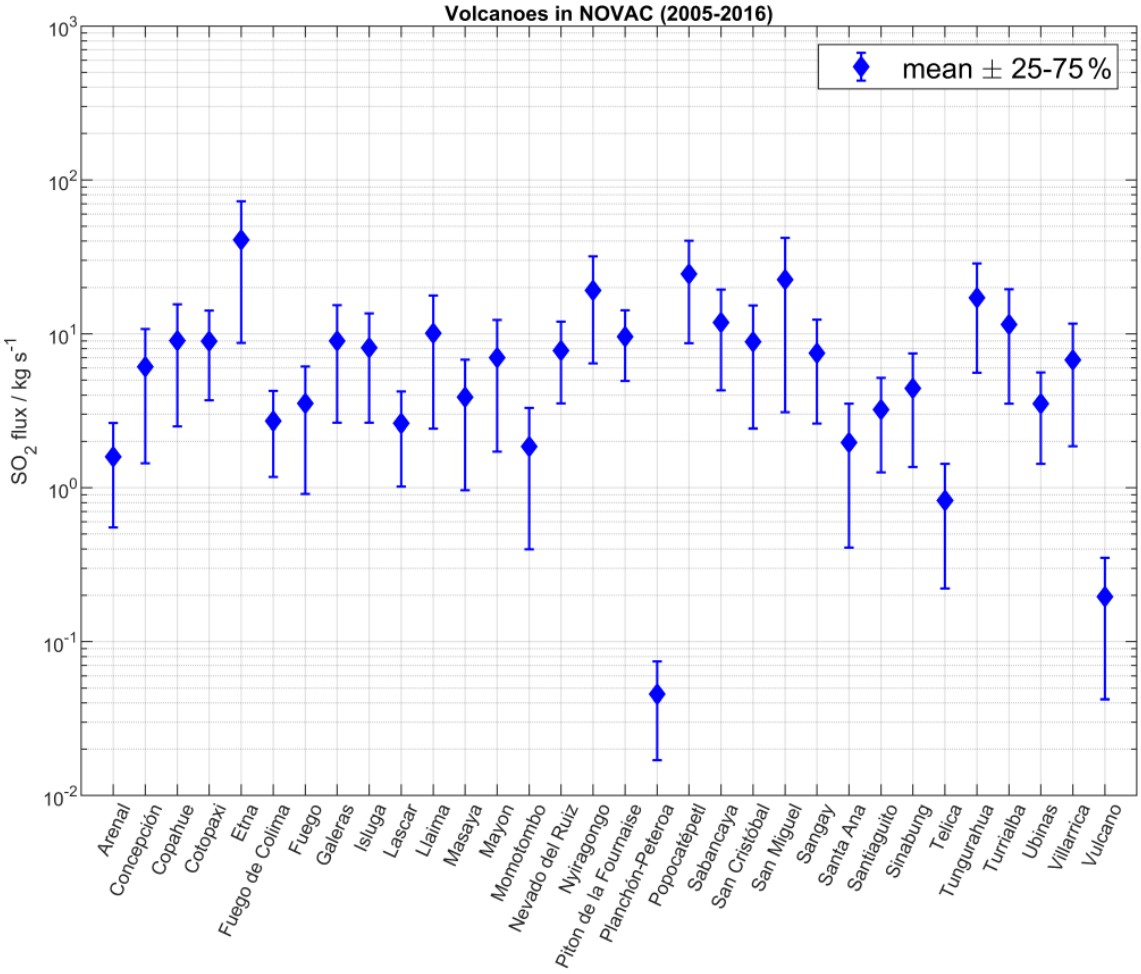

**Figure 4.** Statistics of daily SO$_2$ emission from 32 volcanoes in NOVAC from 1 March 2005 to 31 January 2017, for periods of time when data were being collected and yielding flux values above the detection limit. Blue markers show the average of all measured fluxes for each volcano during this period, and the error bars show the corresponding 25 % and 75 % quantiles.

observation also captures periods of quiescence that may be the reason for lower values.

## 5   Code availability

The NOVAC Post Processing Program and other software used in NOVAC are open-source projects available on GitHub (https://github.com/NOVACProject/, last access: TS36).

## 6   Data availability

More information about NOVAC can be found at the website https://novac-community.org/ (last access: TS37). Raw data from NOVAC are accessible by request to the local observatory responsible for the measurements. The dataset obtained for this study can be accessed, free of charge, through a dedicated website (https://novac.chalmers.se/, last access: TS38).

The datasets of individual volcanoes can be accessed through the DOI links provided in Table 1. Updates of the time series, addition of new volcanoes, and release of data versions resulting from improved analysis are planned.

## 7   Conclusions and outlook

In this study, we report the results of post-processing of SO$_2$ mass emission rate measurements at 32 volcanoes of the NO-VAC network during 2005–2017. This is, to our knowledge, the densest ($\sim$ 10–50 measurements per volcano per day for up to 12 years, 32 volcanoes) database of volcanic degassing obtained by a standardized method. Since the ScanDOAS method is subject to multiple and potentially large sources of uncertainty, considerable attention has been given to the selection of high-quality measurements on which to base the reported statistics.

Independent studies (e.g. Stoiber and Jepsen, 1983 TS39; Krueger et al., 1995; Halmer et al., 2002; Andres and Kas-

**(a)**

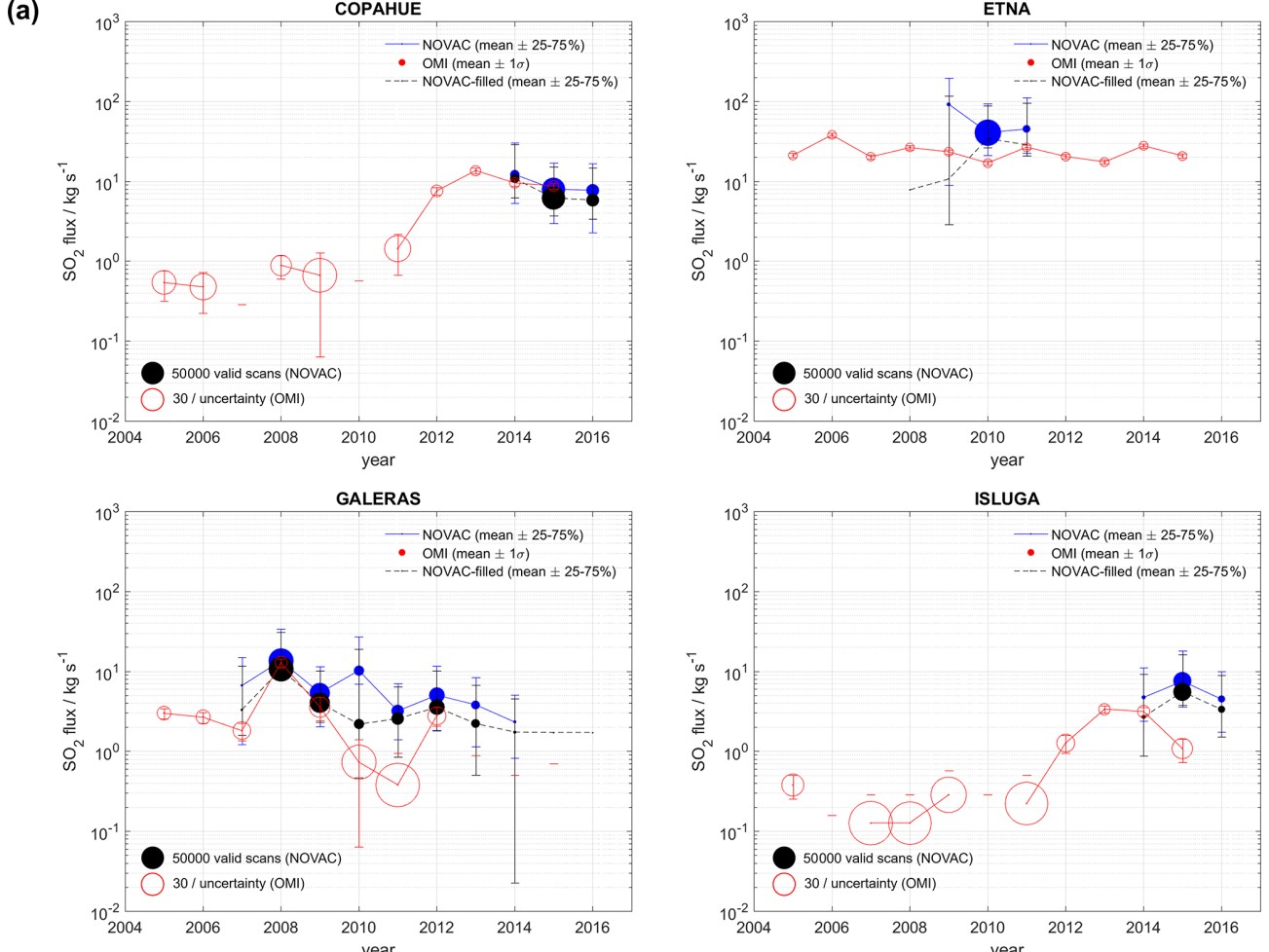

**Figure 5.**

gnoc, 1998; Carn et al., 2017) have demonstrated over the years that passive degassing dominates, in time and magnitude, the time-averaged global volcanic emission. At the same time, this component of volcanic emission can produce a persistent impact on local, regional, or global scales and potentially affect the climate system. Observational limitations have hindered quantifying the magnitude and variability of global volcanic degassing in the past at the level of detail obtained by a global ground-based network like NOVAC. This database will therefore represent an important contribution to global emission inventories, which are typically based on sporadic and short-term investigations of emission during periods of heightened activity. Moreover, the results from measurements in NOVAC complement the observations from satellite platforms, which on an operational basis during the past decade were more suited for quantification of explosive degassing.

The measurements performed in NOVAC provide more information than the gas emission rate of SO$_2$. First, spectroscopic analysis of the data can be used for retrieving the

abundances of other species, as has been proven most systematically for the case of BrO (Lübcke et al., 2014; Dinger et al., 2018; Warnach et al., 2019). Second, in principle all the variables involved in the calculation of the mass flow rate can be obtained from the measurements (plume location, dimensions, and even velocity), which are valuable for modelling studies of volcanic activity and risk, environmental impact, or atmospheric transport.

An important finding of long-term measurements, as in the case presented here, is the empirical distribution of degassing for individual volcanoes. It has often been assumed that the emission of individual volcanoes exhibits a typical value and a symmetric distribution, and global estimates are computed by assuming a skewed distribution for the global volcanic emission. We have found that individual volcanoes, notwithstanding their completely different volcanological characteristics or states of activity, conform to a distribution that looks nearly symmetrical in logarithmic units. This is to say, the logarithm of the emission rate tends to converge to a central value, and the bulk emission is composed of a vast major-

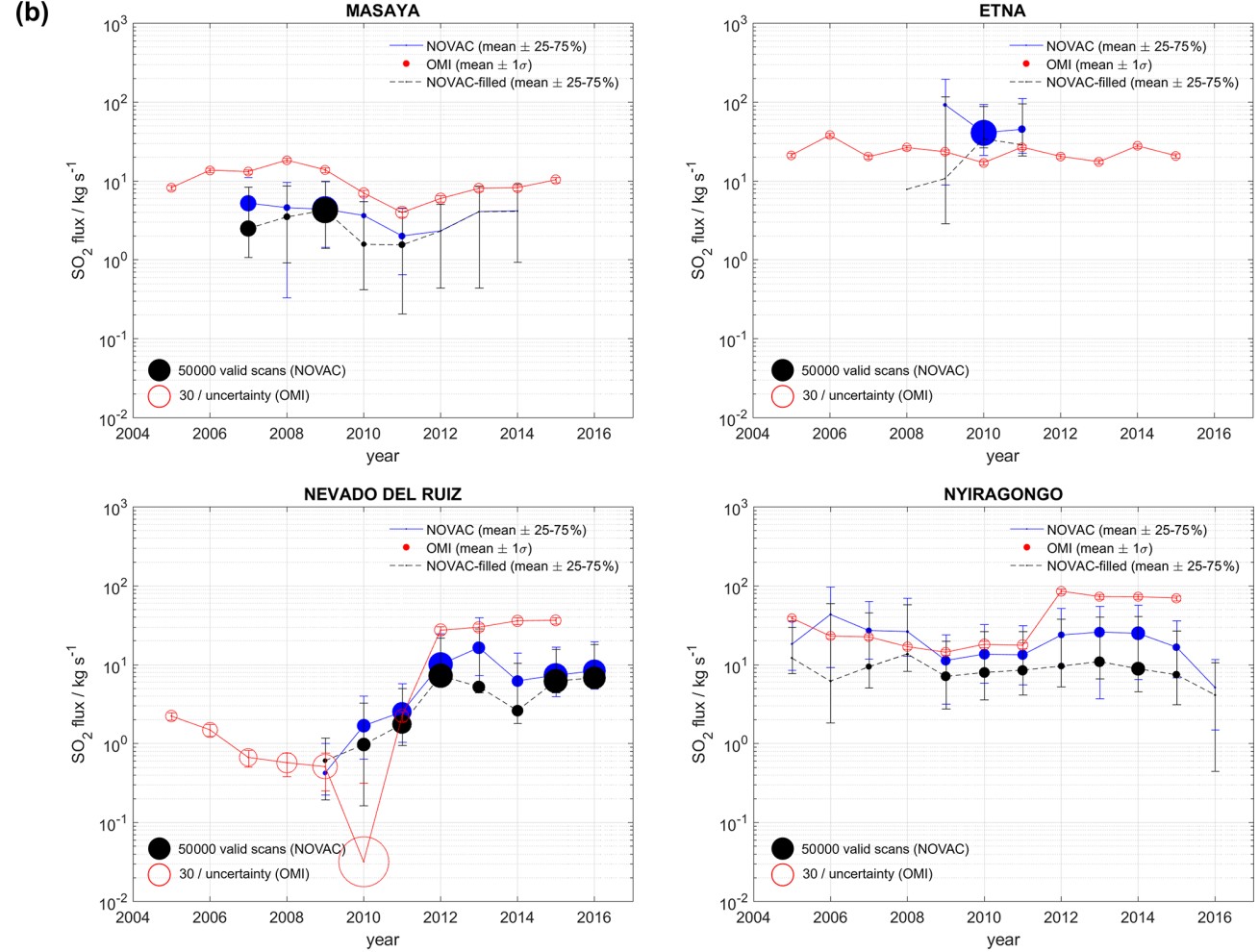

**Figure 5.**

ity of low-emission events and a few but significant high-emission events. It is therefore adequate to typify volcanoes' gas emission in these units, akin to the volcanic sulfur index (VSI) proposed by Schnetzler et al. (1997), which is based on the total emitted sulfur for eruptions rather than the emission rate in general.

We would also like to point out that our standardized processing of data from all volcanoes in the network may not necessarily be the best strategy with regards to obtaining more continuous time series of gas emission. The procedure presented here applies very strict criteria to validate individual SO₂ flux measurements. These criteria make sure that all assumptions behind the method of scanning the plume to derive the flux are fulfilled. However, for volcanoes like Piton de la Fournaise or Vulcano, the result of applying these criteria is a drastic reduction in the number of valid measurements. The reason is that most recent eruptions of Piton de la Fournaise have taken place in flank vents located inside the Enclos Fouqué caldera at altitudes below the location of the NOVAC stations. If the plume does not reach high enough al-

titude above the stations, the coverage will not be complete. Still, measurements will be able to pinpoint the location of the plume and under some circumstances also deliver quantitative information about the flux (e.g. adopting a different method of integration of gas column densities). Gas emission is minimal during inter-eruptive periods (Ref TS40). For Vulcano, the dynamics of degassing is different and characterized by weak passive emission. Even if the emission is detectable by the instruments, winds should be strong and stable enough to produce a plume that fulfils the requirements for scanning. Certain criteria, such as the number of spectra with valid gas detection in the same scan, must be relaxed to obtain a gas flux datum, with a detection of SO₂ down to < 10 t/d (Vita et al., 2012; Granieri et al., 2017). On the other extreme we find episodes when the plume is too wide to completely overcast the scanning path of the instruments. This could happen for volcanoes with strong emission and relatively low altitude difference between the plume and the station. For example, for Nevado del Ruiz, some stations are placed high in the flanks, and due to strong activity of

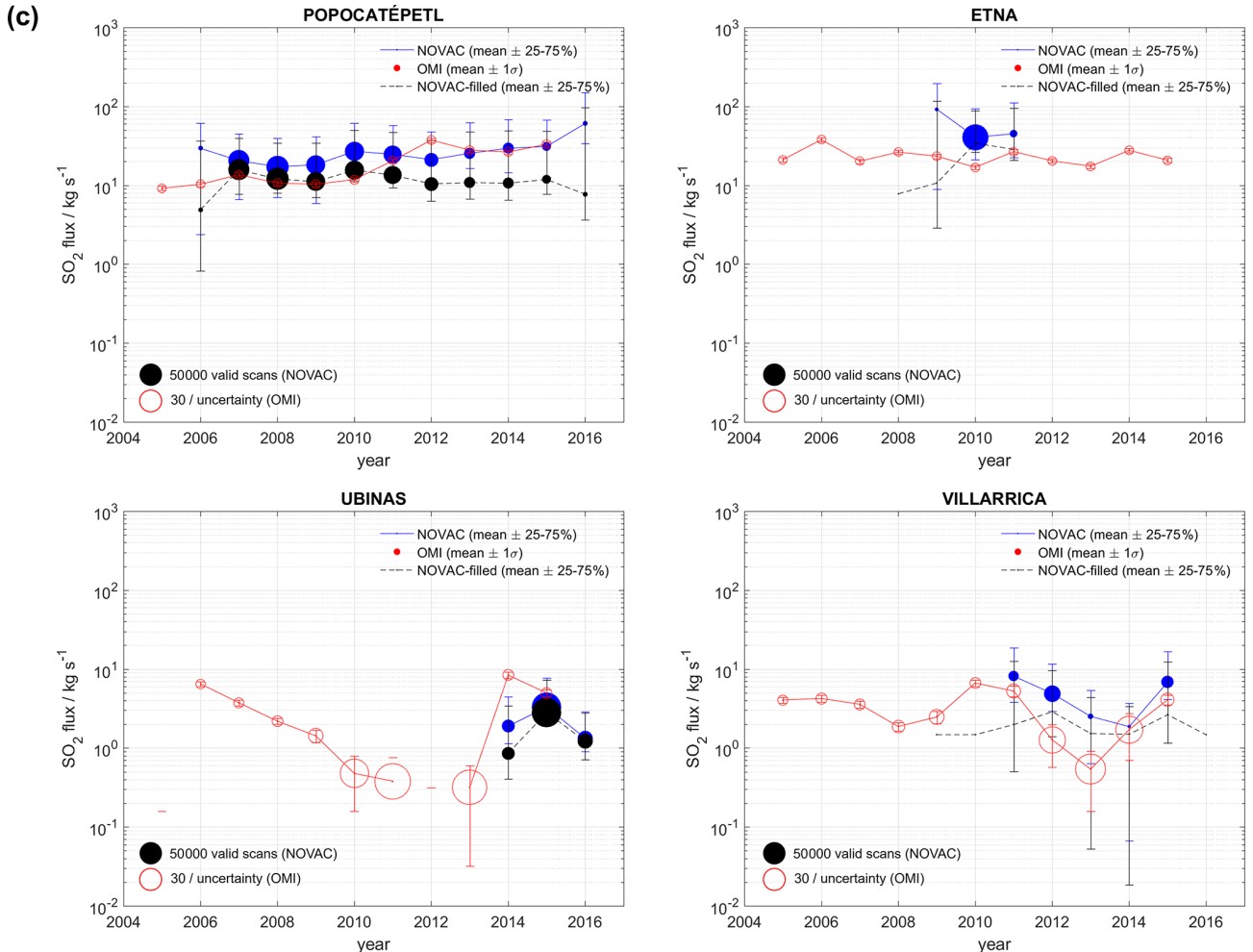

**Figure 5.** Time series of annual SO$_2$ emission from 12 volcanoes in NOVAC for which corresponding results from OMI are available for the period 2005–2016. Dots show the annual mean emission rate with size linearly proportional to the number of valid measurements used for the average. The bars indicate the 25 % and 75 % quantiles of the daily means for each year. The series shown in blue correspond to observed plumes, while in black is the emission adjusted for periods of low degassing, when no plumes were observed (see text for details). The series in red is the mean annual emission rate obtained from OMI measurements, with the size proportional to the precision (reciprocal of uncertainty) of the estimation and bars showing ±1 standard deviation as reported by Carn et al. (2017). **(a)** Data from Copahue, Etna, Galeras, and Isluga. **(b)** Data from Masaya, Mayon, Nevado del Ruiz, and Nyiragongo. **(c)** Data from Popocatépetl, Tungurahua, Ubinas, and Villarrica. For details of the data in numerical format see Supplement S2.

the volcano, the stations will not produce data validated by our method. Using data from more distant stations and novel ways to derive the background SO$_2$ can help to solve this issue (Lübcke et al., 2016). Another example is the Holuhraun eruption, where very extreme conditions in terms of gas column density and low altitude required a different approach to flux estimation (Pfeffer et al., 2018).

There is still a need to improve the characterization of plume transport and radiative transfer effects in remote sensing of volcanic plumes. One of the main advantages of NOVAC is that each improvement in the software or hardware can be easily implemented in the several sites of the network,

and historical data can be re-analysed retrospectively using more advanced algorithms.

Data presented here will be of interest for different applications, but certain considerations are important to mention. First, the reported statistics are considered representative of gas emissions for volcanoes in a state of passive degassing or moderate explosive activity. Emissions resulting from large explosive events may not be properly captured by the near-field, ground-based methods used in NOVAC, since such emissions could either reach several kilometres in the atmosphere, beyond the effective range of the instruments, or they can be too optically thick (due to extreme gas concentrations or aerosols) and thus render the measurements

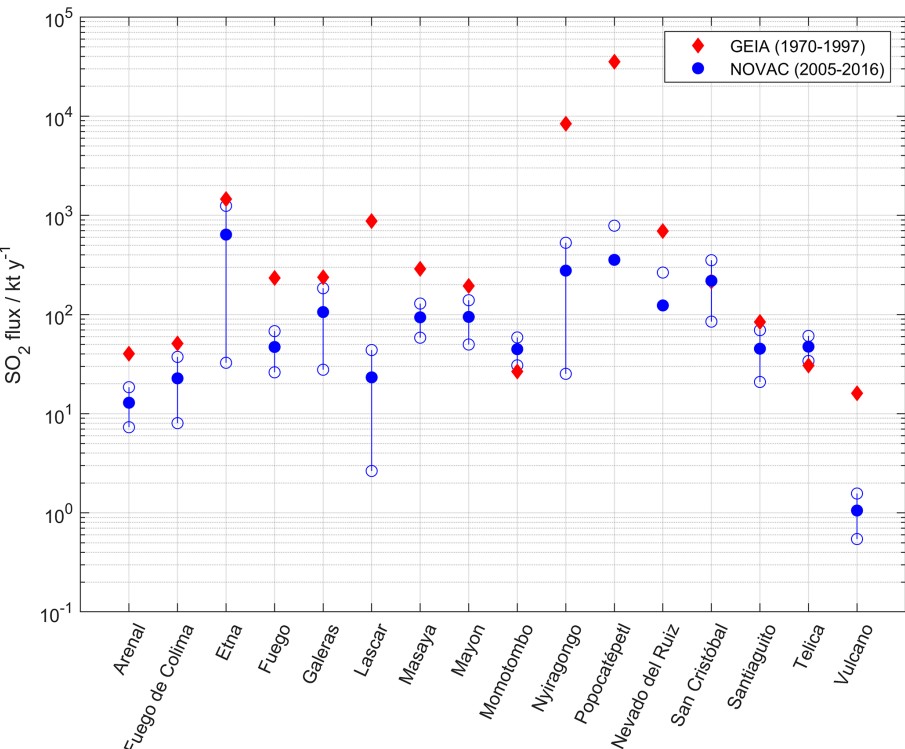

**Figure 6.** Comparison of emission statistics for 16 volcanoes of the NOVAC and GEIA datasets. For NOVAC the simple averages of the annual mean (filled blue dots) and ±1 standard deviations (un-filled blue dots) for the years 2005–2016 are depicted. The values for GEIA are obtained from Andres and Kasgnoc (1998) for passively degassing and sporadically degassing volcanoes only.

inaccurate. Second, our method to derive statistics will be more representative of periods of continuous or frequent degassing. Null-plume detections could be the result of either low or null volcanic emission or due to a lack of observations, caused, for example, by meteorological conditions or not-covered plume directions. Therefore, a simple average of the daily means reported here over a longer period may overestimate the emission, unless null emission is distinguished from null observation, as we propose. On the other hand, the analysis of uncertainty indicates that the effective detections are most likely showing minimum emission values because of the mostly reducing effect of radiation scattering in the retrieved column density values. We think that complementary information to SO₂ flux reported here, namely the statistics of plume location, velocity, dimensions, and general weather conditions, will be valuable not only to interpret the emission patterns but also to assess their impact downwind of the volcano.

NOVAC has expanded since its inception in 2005 and to date includes about a third of all volcanoes which have exhibited degassing detectable from space in the last decade. The network has grown not only from the initiative of volcanological observatories themselves but also from the important support of the USGS–USAID VDAP initiative, especially in Latin America, South East Asia, and Oceania.

Recent initiatives, such as the Deep Carbon Observatory, have built upon the measurements in NOVAC to quantify the global volcanic emission of CO₂, by combining continuous emission rate measurements with punctuated molar ratio measurements (Fischer et al., 2019). Improving the estimates of SO₂ emission will, in this way, result in a better estimate of the emission budgets of other volcanic species. Since volcanic aerosols, to a large extent seeded by primary emission of SO₂, are one of the most important but poorly quantified sources of natural radiative forcing in the climate system (Myhre et al., 2013), there is a need to better quantify their magnitude and location. A global network for the observation of volcanic plumes is of great importance to quantify the magnitude and temporal and spatial variability of volcanic emissions.

**Supplement.** The supplement related to this article is available online at: https://doi.org/10.5194/essd-13-1-2021-supplement.

**Author contributions.** SA conceived the study, analysed data, and wrote the paper with inputs from all authors. BG initiated and coordinated NOVAC and supervised research and database implementation. BG, CK, UP, and THH supervised research for this work at Chalmers University of Technology, USGS VDAP, Heidelberg

University, and GEOMAR, respectively. MJ wrote the spectral analysis software. MK designed the electronics of the instrument. JM supervised development of the database. FA, GA, ChB, NB, ClB, VB, MB, ZC, GC, CJC, VC, FC, MDM, HDG, ADM, DF, GG, HG, NH, SH, SI, PK, PM, FM, CR, AS, GS, BT, FV, GV, FV, and MY supervised research; installed, maintained and operated instruments; and analysed real-time data at their respective observatories.

**Competing interests.** The authors declare that they have no conflict of interest.

**Disclaimer.** Any use of trade, firm, or product names is for descriptive purposes only and does not imply endorsement by the authors or by the governments of their countries, including the United States.

**Special issue statement.** This article is part of the special issue "Surface emissions for atmospheric chemistry and air quality modelling". It is not associated with a conference. TS41

**Acknowledgements.** Initial implementation of the network was funded by the EU FP6 NOVAC project (https://cordis.europa.eu/project/rcn/75513/factsheet/en, last access: TS42). Recent funding for this work was provided by the Alfred P. Sloan Foundation's Deep Carbon Observatory programme (https://deepcarbon.net/, last access: TS43), Chalmers University of Technology, the Swedish National Space Agency (career grant no. 149/18), and the ECMWF CAMS_81 Global and Regional Emissions project. The authors are thankful for the valuable support from the USGS–USAID Volcano Disaster Assistance Program (https://volcanoes.usgs.gov/vdap/, last access: TS44) provided to NOVAC since 2015.

Thanks go to Urban Andersson, Andreas Backström, and Maria Kinger (Chalmers University of Technology) for data management assistance and Grégoire Détrez, Fredrik Gustafsson, Bengt Rydberg, Andreas Skyman, and Martin Zackrisson (MolFlow) for database development. Thanks also go to Sarah Ogburn and Larry Mastin (USGS) for reviewing earlier versions of this paper.

Special thanks go to all colleagues who took part in monitoring activities in the volcano observatories here represented, who have been involved with installation, operation, or data analysis of NOVAC stations.

**Review statement.** This paper was edited by Nellie Elguindi and reviewed by Arlin Krueger and one anonymous referee.

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

## Remarks from the language copy-editor

## Remarks from the typesetter

TS23    Please provide date of last access.

TS24    This reference is not in the reference list.

TS25    Stoiber et al. (1973) is not in the reference list. Do you mean Stoiber and Jepsen (1973)?

TS26    Halmer and Schmincke (2002) is not in the reference list. Do you mean Halmer et al. (2002)?

TS27    Khokhar (2005) is not in the reference list.

TS28    Smekens et al. (2018) is not in the reference list.

TS29    Please provide date of last access.

TS30    Please provide date of last access.

TS31    Please check. Is the format correct with or without a hyphen? This also applies to Mark II below.

TS32    Johansson et al., 2009b is not in the reference list. Do you mean Johansson et al. (2009) or Johansson (2009)?

TS33    Please provide date of last access.

TS34    Please provide date of last access.

TS35    Please provide date of last access.

TS36    Please clarify whether the data set is your own. If yes, please provide a DOI in addition to your GitHub URL, since our reference standard includes DOIs rather than URLs. If you have not yet created a DOI for your data set, please issue a Zenodo DOI (https://help.github.com/en/github/creating-cloning-and-archiving-repositories/referencing-and-citing-content). If the data set is not your own, please inform us accordingly. In any case, please ensure that you include a reference list entry corresponding to the data set including creators, title, and date of last access.

TS37    Please provide date of last access.

TS38    Please provide date of last access.

TS39    Stoiber and Jepsen (1983) is not in the reference list.

TS40    Should a citation be added here?

TS41    Please confirm.

TS42    Please provide date of last access.

TS43    Please provide date of last access.

TS44    Please provide date of last access.

TS45    Please add publisher or repository.

TS46    Please add publisher or repository.

TS47    Please add publisher or repository.

TS48    Please add publisher or repository.

TS49    This reference is not mentioned in the text.

TS50    Please add publisher or repository.

TS51    Please add publisher or repository.

TS52    Please add publisher or repository.

TS53    Please add publisher or repository.

TS54    Please add publisher or repository.

TS55    Please add publisher or repository.

TS56    Please add publisher or repository.

TS57    Please add publisher or repository.

TS58    Please add publisher or repository.

TS59    Please add publisher or repository.

TS60    Please add publisher or repository.

TS61    Please add publisher or repository.

TS62    Please add publisher or repository.

TS63    Khokhar et al. (2005) is not mentioned in the text.

TS64    Please add publisher or repository.

TS65    Please add publisher or repository.

TS66    Please add publisher or repository.

TS67    Please check URL.

TS68    Please add publisher or repository.

TS69    Please add publisher or repository.

TS70    Please add publisher or repository.

TS71    Please add publisher or repository.

TS72    Please add publisher or repository.

TS73    Salerno et al. (2009) is not mentioned in the text.

https://doi.org/10.5194/essd-13-1-2021      Earth Syst. Sci. Data, 13, 1–27, 2021

TS74   Please add publisher or repository.
TS75   Smekens and Gouhier (2018) is not mentioned in the text.
TS76   Stoiber and Jepsen (1973) is not mentioned in the text.
TS77   Please add publisher or repository.
TS78   Please add publisher or repository.
TS79   Please add publisher or repository.
TS80   Please add publisher or repository.
TS81   Please add publisher or repository.