# Peer review of "Synoptic Analysis of a Decade of Daily Measurements of SO2 Emission in the Troposphere from Volcanoes of the Global Ground-Based Network for Observation of Volcanic and Atmospheric Change"

_Earth System Science Data, 2020_

## Referee Comment (RC1) · Arlin Krueger (Referee) · 24 Nov 2020

General Comments:

This is a very well-written, comprehensive paper describing the NOVAC project and database. Furthermore, it puts this project in context in the history of remote sensing observations of volcanic SO2 emissions. It goes well beyond the stated purpose of a "Presentation of an inventory of daily SO2 flux measurements in NOVAC program" after

standardized analysis of spectra and ECMWF reanalysis winds. References are provided to virtually all the important research in this area. It will be an excellent reference for users of the database.

The NOVAC program with routine daily monitoring is an advance over the episodic nature of the previous ground and airborne campaigns in response to explosive eruptions. The authors are to be commended for their extensive efforts to characterize volcanic so2 emissions. While satellites are able to measure even the largest eruptions, passive degassing has remained out of reach, at least until now when TropOMI and the new GEMS geostationary instrument have the ground resolution to detect air pollution levels of SO2. Thus, it will be interesting to see future comparisons.

Ground-based measurements of volcanic SO2 mass are challenging due to changing winds, emission rates, plume size, and cloud and lighting conditions. The dual-beam ScanDOAS instruments are a step forward in automating the data collection. Protocols for observations at the stations are described in detail, as are quality control measures. Steps taken to compensate for changing conditions are described. A post-processing program to assure uniform evaluation of data collected from diverse stations and operators is described.

The sources of error due to the challenging measurement conditions are discussed in some detail. A conservative approach for quality over quantity of observations placed in the database is advocated. Criteria for inclusion of measurements are discussed.

In Sec. 2, I am bothered by treating the word "data" as singular, as in "Data is transferred via a serial port. . ." (line 180, and again in line 189) in the scientific literature. I realize in non-scientific literature "data" can describe a single collection of facts, etc., but I learned to use "datum" for singular measurements and "data" for multiple points.

Sec. 3. Results

Figure 3, consisting of 32 plots of station data, is impossible to read and should be

presented in two or more sub-figures. In Fig. 5 (mis-referenced as fig. 4) the series of annual emissions from 12 volcanoes is compared with OMI data. These 12 panels are just legible on a printed page.

Steps to deal with the problem of calculating long-term emission budgets from irregular sets of data are discussed. The adopted policy of filling missing days by interpolation is understandable but problematic.

Sec. 3.3

The Chalmers data repository website includes site coverage maps, links to the data, and a link to the appropriate GVP page. The documentation seems thorough and includes data use agreements. This is user friendly and seems very well done.

Sec 4.2 Ground-based vs. space-based observations

Even on a log scale the differences between NOVAC and OMI annual emissions are large and the explanations are not convincing. I would think the failure of OMI in separating emissions from two nearby volcanoes would not be important as ground observers certainly will notice emissions from the other volcano, as noted. Limitations of NOVAC data to daytime hours cannot explain factors of 2 - 4 differences even given the high variability of emissions. In line 451ff "whereas OMI could in principle detect. . ." OMI can in fact detect any emissions that have occurred day or night prior to the overpass. However, one has to account for chemical or physical losses in order to calculate instantaneous emission rates and totals. As stated, an in-depth study of discrepancies is certainly needed. This section detracts from the otherwise excellent presentation of the research work. I suggest reworking or removing this section.

Perhaps treating the two datasets as complementary will help explain the differences, as alluded to in the conclusions. It could be that the real value of NOVAC may be in characterizing the continuing low-level background emissions of volcanoes. Any thoughts of validating older satellite data appear to be gone. However, if the suggestion

that low-level background emissions dominate the global volcanic SO2 budget is borne out, then this effort will have been successful.

In the Conclusions section, I am happy to see that Charlie Schnetzler, et al,'s idea of augmenting (or replacing) the qualitative VEI with a quantitative SEI as a measure of eruption sizes has not been totally forgotten (line 504).

The "Author contribution" section acknowledging the roles of team members is certain to be appreciated and is a valuable addition to the paper.

Detailed comments

line 158. "within <10km" is redundant. Use either "within" or "<".

line 237: "hPa", not "Pa".

line 319. "Fig. 5", not "fig. 4".

line 431. "Fig. 5", not "fig. 4".

line 858. "are", not "is".

Figures.

Fig. 2. Two sections of this figure are redundantly labeled "(a)". I assume one of those is actually (b). Please correct this.

Fig. 3. Individual plots are illegible without magnification. Suggest splitting into two or more figures.

---

## Referee Comment (RC2) · Anonymous Referee #2 · 29 Dec 2020

This is a very good and well-written paper. This will be useful during a major eruption to help determine the dynamics of any plumes and could direct air traffic around them.

On line 172, you say "angle of view" but you really mean "field of view"?

In section 4.2, line 431 you reference figure 4 when I think you mean figure 5.

Line 437, you should not include Sangay and Sinabung that you are providing new

information for. NASA's Global Sulfur Dioxide Monitoring pages have been monitoring those volcanoes since 2004.

On Figure 1, the red triangles can override the yellow squares. Maybe make the triangles a little smaller.

Figure 2 has 2 part a)s.

In the Introduction you reference Sparks, et al., 2014, but I don't see it. Maybe 2012.

Some references are out of sequence.

---

## Author Response (AR2)

**Synoptic Analysis of a Decade of Daily Measurements of SO₂ Emission in the Troposphere from Volcanoes of the Global Ground-Based Network for Observation of Volcanic and Atmospheric Change**

Santiago Arellano [1], Bo Galle [1], Fredy Apaza [2], Geoffroy Avard [3], Charlotte Barrington [4], Nicole Bobrowski [5], Claudia Bucarey [6], Viviana Burbano [7(+)], Mike Burton [8,a], Zoraida Chacón [7], Gustavo Chigna [9], Christian Joseph Clarito [10], Vladimir Conde [1], Fidel Costa [4], Maarten De Moor [3], Hugo Delgado-Granados [11], Andrea Di Muro [12], Deborah Fernandez [10], Gustavo Garzón [7], Hendra Gunawan [13], Nia Haerani [13], Thor H. Hansteen [14], Silvana Hidalgo [15], Salvatore Inguaggiato [8], Mattias Johansson [1], Christoph Kern [16], Manne Kihlman [1], Philippe Kowalski [12], Pablo Masias [2], Francisco Montalvo [17], Joakim Möller [18], Ulrich Platt [5], Claudia Rivera [1,b], Armando Saballos [19], Giuseppe Salerno [8], Benoit Taisne [4], Freddy Vásconez [15], Gabriela Velásquez [6], Fabio Vita [8], Mathieu Yalire [20]

[1] Chalmers University of Technology (Chalmers), Department of Space, Earth and Environment, Sweden
[2] Instituto Geológico, Minero y Metalúrgico (INGEMMET), Peru
[3] Observatorio Vulcanológico y Sismológico de Costa Rica (OVSICORI), Costa Rica
[4] Nanyang Technological University, Earth Observatory of Singapore (EOS), Singapore
[5] Heidelberg University, Institute of Environmental Physics, Germany
[6] Servicio Nacional de Geología y Minería (SERNAGEOMIN), Chile
[7] Servicio Geológico Colombiano (SGC), Colombia
[8] Istituto Nazionale di Geofisica e Vulcanologia (INGV), Italy
[9] Instituto Nacional de Sismología, Vulcanología, Meteorología e Hidrología (INSIVUMEH), Guatemala
[10] Philippine Institute of Volcanology and Seismology (PHIVOLCS), Philippines
[11] Universidad Nacional Autónoma de México (UNAM), Instituto de Geofísica, Mexico
[12] Institut de Physique du Globe de Paris (IPGP), Observatoire Volcanologique du Piton de la Fournaise, France
[13] Center for Volcanology and Geological Hazard Mitigation (CVGHM), Indonesia
[14] GEOMAR Helmholtz Centre for Ocean Research Kiel, Germany
[15] Escuela Politécnica Nacional, Instituto Geofísico (IGEPN), Ecuador
[16] United States Geological Survey, Volcano Disaster Assistance Program (USGS/VDAP), United States
[17] Servicio Nacional de Estudios Territoriales (SNET), El Salvador
[18] Möller Data Workflow Systems AB (MolFlow), Sweden
[19] Instituto Nicaragüense de Estudios Territoriales (INETER), Nicaragua
[20] Observatoire Volcanologique de Goma (OVG), DR Congo
[+] Deceased
[a] Now at University of Manchester, School of Earth, Atmospheric and Environmental Sciences, United Kingdom
[b] Now at Universidad Nacional Autónoma de México, Centro de Ciencias de la Atmósfera, Mexico

*Correspondence to*: Santiago Arellano (santiago.arellano@chalmers.se)

**Changes made to the manuscript**

Thank you very much for editorial advice from ESSD and excellent comments from the two referees of our manuscript. We have followed most of the suggestions made by the referees or presented detailed responses when the suggested changes were not adopted. Please refer to the next sections of this document for these detailed responses.

Overall, we have revised entirely the text and found some minor grammatical errors, besides those pointed out by the referees, which are indicated in the file named "essd-2020-295-manuscript_revised_with_tracked_changes.doc".

We have remade all figures according to specific referees' suggestions and following the standards of ESSD.

We have added a paragraph to the Conclusions, after discussion with the co-authors, which reads:

*"We would also like to point out that our standardized processing of data from all volcanoes in the network may not necessarily be the best strategy with regards to obtaining more continuous time-series of gas emission. The procedure presented here applies very strict criteria to validate individual $SO_2$ flux measurements. These criteria make sure that all assumptions behind the method of scanning the plume to derive the flux are fulfilled. However, for volcanoes like Piton de la Fournaise or Vulcano, the result of applying these criteria is a drastic reduction in the number of valid measurements. The reason is that most recent eruptions of Piton de la Fournaise have taken place in flank vents located inside the Enclos Fouqué caldera at altitudes below the location of the NOVAC stations. If the plume does not reach enough altitude above the stations, the coverage will not be complete. Still, measurements will be able to pinpoint the location of the plume, and under some circumstances also deliver quantitative information of the flux (e.g. adopting a different method of integration of gas column densities). Gas emission is minimal during inter-eruptive periods (Ref). For Vulcano, the dynamics of degassing is different and characterized by weak passive emission. Even if the emission is detectable by the instruments, winds should be strong and stable enough to produce a plume that fulfils the requirements for scanning. Certain criteria, such as the number of spectra with valid gas detection in the same scan, must be relaxed to obtain a gas flux datum, with a detection of $SO_2$ down to <10 t/d (Vita et al., 2012; Granieri et al., 2017). On the other extreme we find episodes when the plume is too wide to completely overcast the scanning path of the instruments. This could happen for volcanoes with strong emission and relatively low altitude difference between the plume and the station. For example, for Nevado del Ruiz, some stations are placed high in the flanks and due to strong activity of the volcano, the stations will not produce data validated by our method. Using data from more distant stations and novel ways to derive the background $SO_2$ can help to solve this issue (Lübcke et al., 2016). Another example is the Holuhraun eruption, which very extreme conditions in terms of gas column density and low altitude required a different approach to flux estimation (Pfeffer et al., 2018)."*

This paragraph seeks to raise attention a possible drawback of "standardization" of data analysis, which we consider also important to point out, even though it was not suggested by the referees.

The list of references has been corrected in order, and mistyped date in one reference. We also added three references to the article and a Disclaimer section, as noted in the file with tracked changes

**Response to Interactive Comment by Referee #1, Dr. Arlin Krueger**

We thank Dr. Arlin Krueger for the thorough and thoughtful review of our manuscript and for his many valuable suggestions.

Dr. Krueger's report provides a balanced and complete description of the content of this lengthy manuscript. His overall highly positive remarks are encouraging. In response to the most critical comments (*in italics*) in Dr. Krueger's report we present the following arguments:

*"In Sec. 2, I am bothered by treating the word "data" as singular, as in "Data is transferred via a serial port. . ." (line 180, and again in line 189) in the scientific literature. I realize in non-scientific literature "data" can describe a single collection of facts, etc., but I learned to use "datum" for singular measurements and "data" for multiple points."*

We thank for pointing out this grammatical error, which is now amended throughout the manuscript.

*"Figure 3, consisting of 32 plots of station data, is impossible to read and should be presented in two or more sub-figures. In Fig. 5 (mis-referenced as fig. 4) the series of annual emissions from 12 volcanoes is compared with OMI data. These 12 panels are just legible on a printed page."*

We agree that the figures need to be improved. We have revised all figures, increased their size, and split Figure 3 into eight figures for better visualization. We have also corrected the mis-referencing of Figure 4. Figure 5 is also split into 3 figures for better visualization. Thanks for pointing out this mistake.

*"Steps to deal with the problem of calculating long-term emission budgets from irregular sets of data are discussed. The adopted policy of filling missing days by interpolation is understandable but problematic."*

We agree that the procedure to attribute periods of null detection of volcanic plumes by the instruments to actual low levels of volcanic outgassing may not be entirely satisfactory, and in this sense "problematic", due to lack of detailed information for all conceivable cases. Although the comment does not elaborate on details, we would like to stress that our procedure is a *proposal* to deal with irregular time series allowing a more reasonable comparison with the satellite-based dataset, which is assumed to have a more complete coverage. This is the simplest unbiased alternative to performing an interpolation between observed data-points: when all conditions are met for detecting a volcanic plume, and yet no emission is observed, the most likely explanation is that the emission was in fact low. This is quite different from assigning an emission equal to the mean-point between the two nearest observed points. Beyond the goal of comparing long-term emission budgets between two instruments, the proposed method could be very useful for monitoring purposes, because contextual information on wind direction and cloud cover allows to differentiate between true low emission of the volcano and other non-volcanic causes for missing the plume (weather, plume direction, instrumental failures). This type of information is crucial to interpret the variations in volcanic activity.

Furthermore, we present both the original and the re-constructed time series resulting from applying this method, as well as an indication of the density of data for NOVAC. The cautious reader can then judge the potential bias that this method may cause, for example if the number of days with observed plumes is too low to serve for a conclusive comparison. We think these are valid arguments for keeping both time-series (and tabulated data) in the manuscript and appendix.

*"Even on a log scale the differences between NOVAC and OMI annual emissions are large and the explanations are not convincing."*

Since this is the more substantial criticism to the manuscript, in the following we attempt to respond each assertion separately.

*"I would think the failure of OMI in separating emissions from two nearby volcanoes would not be important as ground observers certainly will notice emissions from the other volcano, as noted."*

We agree and, as mentioned in the comment, already noted that ground-based observations could help to discriminate between nearby sources. This is the case, for example, at the Virunga volcanoes Nyamuragira and Nyiragongo or the Central American volcanoes San Cristóbal and Telica. Inherent to the method to achieve tropospheric sensitivity from OMI observations, is a loss in spatial resolution of sources located just a few tens of km apart. We do not argue that OMI 'fails' in estimating the aggregated emissions of one source region, but rather that by comparing the two methods, as in the case of Nyiragongo presented in Figure 5, we can now estimate the contribution of each individual source to the aggregated emission. Therefore, the large discrepancy observed after 2011 should not be interpreted as a failure of OMI to quantify emissions from Nyiragongo, but instead demonstrate the value of combining both methods to obtain the emissions from each of the two individual volcanoes. We think that the text explains this well:

[…in the case of two nearby volcanoes, such as Nyiragongo-Nyiamulagira (with a footprint of $13 \times 24$ km$^2$), OMI cannot separate completely the contributions of each source, so they are reported as a complex. In this respect, NOVAC can aid to discriminate between these sources, since the stations are deployed with a focus on Nyiragongo and the finer time resolution allows disentangling contributions, especially during periods of heightened activity at any of them.].

*"Limitations of NOVAC data to daytime hours cannot explain factors of 2 - 4 differences even given the high variability of emissions. In line 451ff "whereas OMI could in principle detect. . ." OMI can in fact detect any emissions that have occurred day or night prior to the overpass. However, one has to account for chemical or physical losses in order to calculate instantaneous emission rates and totals."*

We partially disagree with these assertions. Firstly, differences within the same order of magnitude are expected when we consider the uncertainty distributions for both methods. As discussed in section 2.3, NOVAC has an asymmetric uncertainty distribution skewed towards values lower than the mean with a standard deviation which is typically in the order of 50%. By averaging data for one day, the random component of uncertainty is reduced but the reported mean daily flux still lies within about -30 and +10% of the most probable value (as explained in section 2.3 of the manuscript). To compare with OMI, we further calculate statistics over an entire year (with and without the 'filling' procedure) which results in an unknown, but presumably larger uncertainty for the annual emission estimate. This value is more likely an underestimate than an overestimate of the most probable flux, with an uncertainty of around 50%. If we assume, due to lack of better information, that the daily emission derived from averaging of OMI images is symmetric and within ±50%, then it is perfectly reasonable that differences on the order of a factor of 2-4 are sometimes encountered between the annual emissions derived from the two methods. It is worth-mentioning that the differences are largely dependent on the amount of valid data and that in many cases the agreement between the two methods is remarkably good.

Secondly, we used the qualifier "could in principle", because detection of all emissions within the OMI overpass-interval would be feasible for complete coverage of the source and surroundings, for images uncontaminated by clouds or large column density pixels, and by accounting for 'losses' due to chemical reactions, deposition on ground, adsorption on ash or aerosol surfaces, or dilution in the atmosphere below the detection limit of the sensor. For OMI, not all volcanoes can be observed completely every day. This is first a consequence of the orbit of the Aura satellite that leaves narrow gaps ('stripes'), particularly at low latitudes. Since 2007, OMI spectra have also been affected by the so-called 'row-anomaly' (*The OMI Team, Ozone Monitoring Instrument Data User's Manual*) which changes with time and which can have a significant effect on the retrieval, and therefore data affected by this anomaly are removed resulting in a further degradation of daily coverage. Finally, to produce the OMI dataset that we used in our comparison, Carn et al. (2017) applied

filters for cloud cover above 20%, solar zenith angles below 70°, large SO2 column densities (5-15 DU), and other selection criteria. Together, all of these effects lead to a significant reduction in spatial coverage and thus, filtered in this way, the OMI dataset is far from complete on daily timescales.

*"As stated, an in-depth study of discrepancies is certainly needed. This section detracts from the otherwise excellent presentation of the research work. I suggest reworking or removing this section."*

We respectfully disagree with this assertion. The dataset available from OMI (Carn et al., 2017 and related work) is the most complete complementary record of observations which overlaps in time with the ground-based measurements presented in this study. It is therefore highly desirable to compare the two datasets. We acknowledge the difficulties and complexities in performing a high-quality comparison and validation study and agree with the referee that a more thorough examination is needed than what is possible within the scope of this manuscript, which first and foremost focuses on presenting the ground-based data. However, we feel it is necessary to make a first comparison here in order to point out the complexities, difficulties, and potential pitfalls of such an exercise, lest other researchers with less experience in this field attempt an oversimplified comparison and possibly draw incorrect conclusions from it.

We present, as transparently as possible, all the steps from data acquisition to generation of daily emission statistics. We then produce annual averages for comparison by proposing a method to more properly fill data gaps. Still, we find that each volcano should be treated separately because of different conditions of measurement, level of activity or proximity to nearby sources. We hope that this first comparison will elicit a more thorough study, ideally involving both teams behind the production of the two datasets.

*"Perhaps treating the two datasets as complementary will help explain the differences, as alluded to in the conclusions. It could be that the real value of NOVAC may be in characterizing the continuing low-level background emissions of volcanoes. Any thoughts of validating older satellite data appear to be gone. However, if the suggestion that low-level background emissions dominate the global volcanic SO2 budget is borne out, then this effort will have been successful."*

We agree with this comment, that both datasets are to a large degree complementary. The assertion that low-level background volcanic emissions of $SO_2$ are dominant is not originally ours but was instead determined by global satellite-based measurements. Since the first detection of $SO_2$ emitted during the 1982 El Chichón eruption by our referee and with increasing sensitivity over the years, satellite observations have demonstrated the predominance of passive over eruptive volcanic degassing. We believe NOVAC could play a crucial role in refining this picture. The reason behind validating OMI data only in this study is the availability of a very thorough and comprehensive dataset for this sensor; however, extending this comparison to other sensors such as GOME-2 or OMPS, which have been operational during the same period, would be very interesting. This would allow an assessment of the effects of different spatial resolutions and other instrumental factors in producing time-series of volcanic gas emissions from satellite records.

*"Detailed comments"*

We thank the reviewer for spotting these errors, all of which are now amended.

**Response to Interactive Comment by Referee #2**

We thank Referee #2 for the detailed review of our manuscript and valuable suggestions.

Below are our responses to questions and suggestions posed by Referee #2.

*"This is a very good and well-written paper."*

Thank you!

*"This will be useful during a major eruption to help determine the dynamics of any plumes and could direct air traffic around them."*

Indeed, the technique employed in our network provides real-time detection of the amount and spatial location of $SO_2$ plumes at a few km away from the vent. At such distance it is likely that $SO_2$ is well mixed with the rest of components of the plume, including eventually ash in the case of explosive emissions. Gas emission rate and plume altitude are key pieces of information of the source conditions required to initialize volcanic plume dispersion models. These models, together with additional observations from satellite, are used by Volcanic Ash Advisory Centers (VAACs) set up by the International Civil Aviation Organization (ICAO). These centers are groups of experts responsible for coordinating and distributing information on volcanic ash clouds that may pose a danger to aviation.

Although our paper focuses on presenting historical data, the statistics provided of gas emission rate, plume altitude and velocity, can be used to define most likely scenarios of plume dispersion that could be considered to plan air traffic.

*"On line 172, you say "angle of view" but you really mean "field of view"?"*

Yes, thanks for pointing out this unusual use of the term. We changed it.

*"In section 4.2, line 431 you reference figure 4 when I think you mean figure 5."*

Yes, thanks for pointing out this error, which is now corrected.

*"Line 437, you should not include Sangay and Sinabung that you are providing new information for. NASA's Global Sulfur Dioxide Monitoring pages have been monitoring those volcanoes since 2004."*

We have included Sangay and Sinabung in the list of volcanoes for which there is no previous estimates of gas emission neither in the compilations from Andreas and Kasgnoc (1997) and Carn et al. (2017).

We recognize that NASA's Global Sulfur Dioxide Monitoring service (https://so2.gsfc.nasa.gov/) has global coverage and therefore it has detected emission from these volcanoes. However, emission from the mentioned volcanoes is more recent and was not included in the compilation published in 2017, which has been peer-reviewed. We therefore decided to keep the text unchanged.

*"On Figure 1, the red triangles can override the yellow squares. Maybe make the triangles*

*a little smaller."*

Thanks for the suggestion. We have remade all figures including yours, and Referee #1's comments. It is now possible to distinguish the locations of volcanoes listed in the GVP, OMI and NOVAC databases.

*"Figure 2 has 2 part a)s."*

Thanks! Corrected.

*"In the Introduction you reference Sparks, et al., 2014, but I don't see it. Maybe 2012."*

Thanks! Corrected.

*"Some references are out of sequence."*

Thanks! We have revised the entire list of references and sort them out correctly.